# On Sample Complexity Upper and Lower Bounds for Exact Ranking from Noisy Comparisons

**Wenbo Ren**
Department of Computer Science & Engineering
The Ohio State University
`ren.453@osu.edu`

**Jia Liu**
Department of Computer Science
Iowa State University
`jialiu@iastate.edu`

**Ness B. Shroff**
Department of Electrical & Computer Engineering and Computer Science & Engineering
The Ohio State University
`shroff.11@osu.edu`

## Abstract

This paper studies the problem of finding the exact ranking from noisy comparisons. A noisy comparison over a set of $m$ items produces a noisy outcome about the most preferred item, and reveals some information about the ranking. By repeatedly and adaptively choosing items to compare, we want to fully rank the items with a certain confidence, and use as few comparisons as possible. Different from most previous works, in this paper, we have three main novelties: (i) compared to prior works, our upper bounds (algorithms) and lower bounds on the sample complexity (aka number of comparisons) require the minimal assumptions on the instances, and are not restricted to specific models; (ii) we give lower bounds and upper bounds on instances with *unequal* noise levels; and (iii) this paper aims at the *exact* ranking without knowledge on the instances, while most of the previous works either focus on approximate rankings or study exact ranking but require prior knowledge. We first derive lower bounds for pairwise ranking (i.e., compare two items each time), and then propose (nearly) *optimal* pairwise ranking algorithms. We further make extensions to listwise ranking (i.e., comparing multiple items each time). Numerical results also show our improvements against the state of the art.

## 1 Introduction

**Background and motivation:** Ranking from noisy comparisons has been a canonical problem in the machine learning community, and has found applications in various areas such as social choices [8], web search [9], crowd sourcing [4], and recommendation systems [3]. The main goal of ranking problems is to recover the full or partial rankings of a set of items from noisy comparisons. The items can refer to various things, such as products, movies, pages, and advertisements, and the comparisons refer to tests or queries about the items' strengths or the users' preferences. In this paper, we use words "item", "comparison" and "preference" for simplicity. A comparison involves two (i.e., pairwise) or multiple (i.e., listwise) items, and returns a noisy result about the most preferred one, where "noisy" means that the comparison outcome is random and the returned item may not be the most preferred one. A noisy comparison reveals some information about the ranking of the items. This information can be used to describe users' preferences, which helps applications such as recommendations, decision making, and advertising, etc. One example is e-commerce: A user's click or purchase of a product (but not others) is based on a noisy (due to the lack of full information) comparison between several similar products, and one can rank the products based

on the noisy outcomes of the clicks or the purchases to give better recommendations. Due to the wide applications, in this paper, we do not focus on specific applications and regard comparisons as black-box procedures.

This paper studies the *active* (or adaptive) ranking, where the learner adaptively chooses items to compare based on previous comparison results, and returns a ranking when having enough confidence. Previous works [4, 28] have shown that, compared to non-adaptive ranking, active ranking can significantly reduce the number of comparisons needed and achieve a similar confidence or accuracy. In some applications such as news apps, the servers are able to adaptively choose news to present to the users and collect feedbacks, by which they can learn the users' preferences in shorter time compared to non-adaptive methods and may provide better user experience.

We focus on the active *full* ranking problem, that is, to find the *exact* full ranking with a certain confidence level by adaptively choosing the items to compare, and try to use as few comparisons as possible. The comparisons can be either pairwise (i.e., comparing two items each time) or listwise (i.e., comparing more than two items each time). We are interested in the upper and lower bounds on the sample complexity (aka number of comparisons needed). We are also interested in understanding whether using listwise comparisons can reduce the sample complexity.

**Models and problem statement:** There are $n$ items in total, indexed by $1, 2, 3, ..., n$. Given a comparison over a set $S$, each item $i \in S$ has $p_{i,S}$ probability to be returned as the most preferred one (also referred to as $i$ "wins" this comparison), and when a tie happens, we randomly assign one item as the winner, which makes $\sum_{i \in S} p_{i,S} = 1$ for all set $S \subset [n]$. When $|S| = 2$, we say this comparison is pairwise, and when $|S| > 2$, we say listwise. In this paper, a comparison is said to be $m$-wise if it involves exactly $m$ items (i.e., $|S| = m$). For $m = 2$ and a two-sized set $S = \{i, j\}$, to simplify notation, we define $p_{i,j} := p_{i,S}$ and $p_{j,i} := p_{j,S}$.

*Assumptions.* In this paper, we make the following assumptions: **A1)** Comparisons are independent across items, sets, and time. We note that the assumption of independence is common in the this area (e.g., [10, 11, 12, 15, 16, 22, 31, 32, 34, 35]). **A2)** There is a unique permutation $(r_1, r_2, ..., r_n)$ of $[n]$ [1] such that $r_1 \succ r_2 \succ \cdots \succ r_n$, where $i \succ j$ denotes that $i$ ranks higher than $j$ (i.e., $i$ is more preferred than $j$). We refer to this unique permutation as the *true ranking* or *exact ranking*, and our goal is to recover the true ranking; **A3)** For any set $S$ and item $i \in S$, if $i$ ranks higher than all other items $k$ of $S$, then $p_{i,S} > p_{k,S}$. For pairwise comparisons, A3 states that $i \succ j$ if and only if $p_{i,j} > 1/2$. We note that for pairwise comparisons, A3 can be viewed as the weak stochastic transitivity [33]. The three assumptions are necessary to make the exact ranking (i.e., finding the unique true ranking) problem meaningful, and thus, we say our assumptions are minimal. Except for the above three assumptions, we do *not* assume any prior knowledge of the $p_{i,S}$ values. We note that any comparison model can be fully described by the comparison probabilities $(p_{i,S} : i \in S, S \subset [n])$.

We further define some notations. Two items $i$ and $j$ are said to be *adjacent* if in the true ranking, there does *not* exist an item $k$ such that $i \succ k \succ j$ or $j \succ k \succ i$. For all items $i$ and $j$ in $[n]$, define $\Delta_{i,j} := |p_{i,j} - 1/2|$, $\Delta_i := \min_{j \neq i} \Delta_{i,j}$, and $\tilde{\Delta}_i := \min\{\Delta_{i,j} : i \text{ and } j \text{ are adjacent}\}$. We adopt the notion of strong stochastic transitivity (SST) [11]: for all items $i$, $j$, and $k$ satisfying $i \succ j \succ k$, it holds that $p_{i,k} \geq \max\{p_{i,j}, p_{j,k}\}$. Under the SST condition, we have $\Delta_i = \tilde{\Delta}_i$ for all items $i$. We note that this paper is not restricted to the SST condition. Pairwise (listwise) ranking refers to ranking from pairwise (listwise) comparisons. In this paper, $f \preceq g$ means $f = O(g)$, $f \succeq g$ means $f = \Omega(g)$, and $f \simeq g$ means $f = \Theta(g)$. The meanings of $O(\cdot)$, $o(\cdot)$, $\Omega(\cdot)$, $\omega(\cdot)$, and $\Theta(\cdot)$ are standard in the sense of Bachmann-Landau notation with respect to $(n, \delta^{-1}, \epsilon^{-1}, \Delta^{-1}, \eta^{-1}, (\Delta_{i,j}^{-1}, i \neq j))$. For any $a, b \in \mathbb{R}$, define $a \wedge b := \min\{a, b\}$ and $a \vee b := \max\{a, b\}$.

**Problem** (Exact ranking). *Given $\delta \in (0, 1/2)$ and $n$ items, one wants to determine the true ranking with probability at least $1 - \delta$ by adaptively choosing sets of items to compare.*

**Definition 1** ($\delta$-correct algorithms). *An algorithm is said to be $\delta$-correct for a problem if for any input instance of this problem, it, with probability at least $1 - \delta$, returns a correct result in finite time.*

**Main results:** First, for $\delta$-correct pairwise ranking algorithms with no prior knowledge of the instances, we derive a sample-complexity lower bound of the form $\Omega(\sum_{i \in [n]} \Delta_i^{-2}(\log \log \Delta_i^{-1} + \log(n/\delta)))$ [2], which is shown to be *tight* (up to constant factors) under SST and some mild conditions.

Second, for pairwise and listwise ranking under the multinomial logit (MNL) model, we derive a model-specific lower bound, which is *tight* (up to constant factors) under some mild conditions, and shows that in the worst case, the listwise lower bound is no lower than the pairwise one.

Third, we propose a pairwise ranking algorithm that requires no prior information and minimal assumptions on the instances, and its sample-complexity upper bound *matches* the lower bounds proved in this paper under the SST condition and some mild conditions, implying that both upper and lower bounds are optimal.

## 2    Related works

Dating back to 1994, the authors of [14] studied the noisy ranking under the strict constraint that $p_{i,j} \geq 1/2 + \Delta$ for any $i \succ j$, where $\Delta > 0$ is priorly *known*. They showed that any $\delta$-correct algorithm needs $\Theta(n\Delta^{-2} \log(n/\delta))$ comparisons for the worst instances. However, in some cases, it is impossible to either assume the knowledge of $\Delta$ or require $p_{i,j} \geq 1/2 + \Delta$ for any $i \succ j$. Also, their bounds only depend on the minimal gap $\Delta$ but not $\Delta_{i,j}$'s or $\Delta_i$'s, and hence is not tight in most cases. In contrast, our algorithms require *no knowledge* on the gaps (i.e., $\Delta_{i,j}$'s), and we establish sample-complexity lower bounds and upper bounds that base on *unequal* gaps, which can be much tighter when $\Delta_i$'s vary a lot.

Another line of research is to explore the *probably approximately correct* (PAC) ranking (which aims at finding a permutation $(r_1, r_2, ..., r_n)$ of $[n]$ such that $p_{r_i,r_j} \geq 1/2 - \epsilon$ for all $i < j$, where $\epsilon > 0$ is a given error tolerance) under various pairwise comparison models [10, 11, 12, 29, 31, 32, 35]. When $\epsilon > 0$, the PAC ranking may not be unique. The authors of [10, 11, 12] proposed algorithms with $O(n\epsilon^{-2} \log(n/\delta))$ upper bound for PAC ranking with tolerance $\epsilon > 0$ under SST and the stochastic triangle inequality[3] (STI). When $\epsilon$ goes to zero, the PAC ranking reduces to the true ranking. However, when $\epsilon > 0$, we still need some prior knowledge on $(p_{i,j} : i, j \in [n])$ to get the true ranking, as we need to know a lower bound of the values of $\Delta_{i,j}$ to ensure that the PAC ranking equals to the unique true ranking. When $\epsilon = 0$, the algorithms in [10, 11, 12] do not work. Prior to these works, the authors of [35] also studied the PAC ranking. In their work, with $\epsilon = 0$, the unique true ranking can be found by $O(n \log n \cdot \max_{i \in [n]} \{\Delta_i^{-2} \log(n\delta^{-1}\Delta_i^{-1})\})$ comparisons, which is higher than the lower bound and upper bound proved in this paper by at least a log factor.

In contrast, this paper is focused on recovering the *unique* true (exact) ranking, and there are three major motivations. First, in some applications, we prefer to find the exact order, especially in "winner-takes-all" situations. For example, when predicting the winner of an election, we prefer to get the exact result but not the PAC one, as only a few votes can completely change the result. Second, analyzing the exact ranking can help us better understand the instance-wise upper and lower bounds about the ranking problems, while the bounds of PAC ranking (e.g., in [10, 11, 12]) may only work for the worst cases. Third, exact ranking algorithms may better exploit the large gaps (e.g., $\Delta_i$'s) to achieve lower sample complexities. In fact, when finding the PAC ranking, we can perform the exact ranking algorithm and the PAC ranking algorithm parallelly, and return a ranking whenever one of them returns. By this, when $\epsilon$ is large, we can benefit from the PAC upper bounds that depend on $\epsilon^{-2}$, and when $\epsilon$ is small, we can benefit from the exact ranking bounds that depend on $\Delta_i^{-2}$.

There are also other interesting active ranking works. Authors of [15, 16, 22, 34] studied active ranking under the Borda-Score model, where the Borda-Score of item $i$ is defined as $\frac{1}{n-1} \sum_{j \neq i} p_{i,j}$. We note that the Borda-Score model does not satisfy A2 and A3 and is not comparable with the model in this paper. There are also many works on best item(s) selection, including [1, 5, 7, 19, 26, 27, 32], which are less related to this paper.

## 3    Lower bound analysis

### 3.1    Generic lower bound for $\delta$-correct algorithms

In this subsection, we establish a sample-complexity lower bound for pairwise ranking. The lower bound is for $\delta$-correct algorithms, which have performance guarantee for all input instances. There are algorithms that work faster than our lower bound but only return correct results with $1 - \delta$

confidence for a restricted class of instances, which is discussed in Section A.1 of supplementary material. Theorem 2 states the lower bound, and its full proof is provided in supplementary material. Here we remind that $\tilde{\Delta}_i := \min\{\Delta_{i,j} : i \text{ and } j \text{ are adjaent}\}$.

**Theorem 2** (Lower bound for pairwise ranking). *Given $\delta \in (0, 1/12)$ and an instance $\mathcal{I}$ with $n$ items, then the number of comparisons used by a $\delta$-correct algorithm $\mathcal{A}$ with no prior knowledge about the gaps of $\mathcal{I}$ is lower bounded by*

$$\Omega\big(\sum_{i\in[n]}[\tilde{\Delta}_i^{-2}(\log\log\tilde{\Delta}_i^{-1} + \log(1/\delta))] + \min\{\sum_{i\in[n]}\tilde{\Delta}_i^{-2}\log(1/x_i) : \sum_{i\in[n]}x_i \le 1\}\big). \quad (1)$$

*If $\delta \preceq 1/poly(n)^4$, or $\max_{i,j\in[n]}\{\tilde{\Delta}_i/\tilde{\Delta}_j\} \preceq n^{1/2-p}$ for some constant $p > 0$, then the lower bound becomes*

$$\Omega\big(\sum_{i\in[n]}\tilde{\Delta}_i^{-2}(\log\log\tilde{\Delta}_i^{-1} + \log(n/\delta))\big). \quad (2)$$

**Remark:** (i) When the instance satisfies the SST condition (the algorithm does not need to know this information), the bound in Eq. (2) is tight (up to a constant factor) under the given condition, which will be shown in Theorem 12 later. (ii) The lower bound in Eq. (1) implies an $n \log n$ term in $\min\{\cdot\}$, which can be checked by the convexity of $\log(1/x_i)$ and Jensen's inequality, which yields $\sum_{i\in[n]}\log(1/x_i) \ge n\log(n/\sum_{i\in[n]}x_i) \ge n\log n$. (iii) The lower bound in (2) may not hold if the required conditions do not hold, which will be discussed in Section A.2 of supplementary material.

***Proof sketch of Theorem 2.*** Due to space limitation, we outline the basic idea of the proof here and refer readers to supplementary material for details. Our first step is to use the results in [13, 18, 25] to establish a lower bound for ranking two items. Then, it seems straightforward that the lower bound for ranking $n$ items can be obtained by summing up the lower bounds for ranking $\{q_1, q_2\}$, $\{q_2, q_3\},...,\{q_{n-1}, q_n\}$, where $q_1 \succ q_2 \succ \cdots \succ q_n$ is the true ranking. However, Note that to rank $q_i$ and $q_j$, there may be an algorithm that compares $q_i$ and $q_j$ with other items like $q_k$, and uses the comparison outcomes over $\{q_i, q_k\}$ and $\{q_j, q_k\}$ to determine the order of $q_i$ and $q_j$. Since it is unclear to what degree comparing $q_i$ and $q_j$ with other items can help to rank $q_i$ and $q_j$, the lower bound for ranking $n$ items cannot be simply obtained by summing up the lower bounds for ranking 2 items. To overcome this challenge, our strategy is to construct two problems: $\mathcal{P}_1$ and $\mathcal{P}_2$ with decreasing influence of this type of comparisons. Then, we prove that $\mathcal{P}_1$ reduces to exact ranking and $\mathcal{P}_2$ reduces to $\mathcal{P}_1$. Third, we prove a lower bound on $\delta$-correct algorithms for solving $\mathcal{P}_2$, which yields a lower bound for exact ranking. Finally, we use this lower bound to get the desired lower bounds in Eq. (1) and Eq. (2). $\qquad\square$

## 3.2 Model-specific lower bound

In Section 3.1, we provide a lower bound for $\delta$-correct algorithms that do not require any knowledge of the instances except assumptions A1 to A3. However, in some applications, people may focus on a specific model, and hence, the algorithm may have further knowledge about the instances, such as the model's restrictions. Hence, the lower bound in Theorem 2 may not be applicable any more[5].

In this paper, we derive a model-specific lower bound for the MNL model. The MNL model can be applied to both pairwise and listwise comparisons. For pairwise comparisons, the MNL model is mathematically equivalent to the Bradley-Terry-Luce (BTL) model [24] and the Plackett-Luce (PL) model [35]. There have been many prior works that focus on active ranking based on this model (e.g., [5, 6, 7, 15, 19, 27, 31, 35]).

Under the MNL model, each item holds a real number representing the users' preference over this item, where the larger the number, the more preferred the item. Specifically, each item $i$ holds a parameter $\gamma_i \in \mathbb{R}$ such that for any set $S$ containing $i$, $p_{i,S} = \exp(\gamma_i)/\sum_{j\in S}\exp(\gamma_j)$. To simplify notation, we let $\theta_i = \exp(\gamma_i)$, hence, $p_{i,S} = \theta_i/\sum_{j\in S}\theta_j$. We name $\theta_i$ as the *preference score* of item $i$. We define $\Delta_{i,j} := |p_{i,j} - 1/2|$, $\Delta_i := \min_{j\neq i}\Delta_{i,j}$, and we have $\tilde{\Delta}_i = \Delta_i$, i.e., the MNL model satisfies the SST condition.

**Theorem 3.** *[Lower bound for the MNL model] Let $\delta \in (0, 1/12)$ and given a $\delta$-correct algorithm $\mathcal{A}$ with the knowledge that the input instances satisfy the MNL model, let $N_{\mathcal{A}}$ be the number of comparisons conducted by $\mathcal{A}$, then $\mathbb{E}[N_{\mathcal{A}}]$ is lower bounded by Eq. (1) with a different hidden constant factor. When $\delta \preceq 1/poly(n)$ or $\max_{i,j \in [n]}\{\Delta_i/\Delta_j\} \preceq n^{1/2-p}$ for some constant $p > 0$, the sample complexity is lower bounded by Eq. (2) with a different hidden constant factor.*

***Proof sketch.*** We prove this theorem by Lemmas 4, 5 and 6, which could be of independent interest.

Suppose that there are two coins with unknown *head probabilities* (the probability that a toss produces a head) $\lambda$ and $\mu$, respectively, and we want to find the more biased one (i.e., the one with the larger head probability). Lemma 4 states a lower bound on the number of heads or tails generated for finding the more biased coin, which works even if $\lambda$ and $\mu$ go to 0. This is in contrast to the lower bounds on the number of tosses given by previous works [18, 21, 25], which go to infinity as $\lambda$ and $\mu$ go to 0.

**Lemma 4** (Lower bound on number of heads). *Let $\lambda + \mu \leq 1$, $\Delta := |\lambda/(\lambda + \mu) - 1/2|$, and $\delta \in (0, 1/2)$ be given. To find the more biased coin with probability $1 - \delta$, any $\delta$-correct algorithm for this problem produces $\Omega(\Delta^{-2}(\log \log \Delta^{-1} + \log \delta^{-1}))$ heads in expectation.*

Now we consider $n$ coins $C_1, C_2, ..., C_n$ with mean rewards $\mu_1, \mu_2, ..., \mu_n$, respectively, where for any $i \in [n]$, $\theta_i/\mu_i = c$ for some constant $c > 0$. Define the gaps of coins $\Delta_{i,j}^c := |\mu_i/(\mu_i + \mu_j) - 1/2|$, and $\Delta_i^c := \min_{j \neq i} \Delta_{i,j}^c$. We can check that for all $i$ and $j$, $\Delta_{i,j}^c = \Delta_{i,j}$, and $\Delta_i = \tilde{\Delta}_i = \Delta_i^c$.

**Lemma 5** (Lower bound for arranging coins). *For $\delta < 1/12$, to arrange these coins in ascending order of head probabilities, the number of heads generated by any $\delta$-correct algorithm is lower bounded by Eq. (1) with a (possibly) different hidden constant factor.*

The next lemma shows that any algorithm that solves a ranking problem under the MNL model can be transformed to solve the pure exploration multi-armed bandit (PEMAB) problem with Bernoulli rewards(e.g., [18, 20, 30]). Previous works [1, 15, 16] have shown that certain types of pairwise ranking problems (e.g., Borda-Score ranking) can also be transformed to PEMAB problems. But in this paper, we make a *reverse connection* that bridges these two classes of problems, which may be of independent interest. We note that in our prior work [29], we proved a similar result.

**Lemma 6** (Reducing PEMAB problems to ranking). *If there is a $\delta$-correct algorithm that correctly ranks $[n]$ with probability $1-\delta$ by $M$ expected number of comparisons, then we can construct another $\delta$-correct algorithm that correctly arranges the coins $C_1, C_2, ..., C_n$ in the order of ascending head probabilities with probability $1 - \delta$ and produces $M$ heads in expectation.*

The theorem follows by Lemmas 5 and 6. A full proof can be found in supplementary material. $\square$

### 3.3 Discussions on listwise ranking

A listwise comparison compares $m$ ($m > 2$) items and returns a noisy result about the most preferred item. It is an interesting question whether exact ranking from listwise comparisons requires less comparisons. The answer is "It depends." When every comparison returns the most preferred item with high probability (w.h.p.)[6], then, by conducting $m$-wise comparisons, the number of comparisons needed for exact ranking is $\Theta(n \log_m n)$, i.e., there is a $\log m$ reduction, which is stated in Proposition 7. The proof can be found in supplementary material.

**Proposition 7** (Listwise ranking with negligible noises). *If all comparisons are correct w.h.p., to exactly rank $n$ items w.h.p. by using $m$-wise comparisons, $\Theta(n \log_m n)$ comparisons are needed.*

In general, when the "w.h.p. condition" is violated, listwise ranking does not necessarily require less comparisons than pairwise ranking (in order sense). Here, we give an example. For more general models, it remains an open problem to identify the theoretical limits, which is left for future studies.

**Theorem 8.** *Under the MNL model, given $n$ items with preference scores $\theta_1, \theta_2, ..., \theta_n$ and $\Delta_{i,j} := |\theta_i/(\theta_i + \theta_j) - 1/2|$, $\tilde{\Delta}_i = \Delta_i := \min_{j \neq i} \Delta_{i,j}$, to correctly rank these $n$ items with probability $1 - \delta$, even with $m$-wise comparisons for all $m \in \{2, 3, ..., n\}$, the lower bound is the same as the pairwise ranking (i.e., Theorem 3) with (possibly) different hidden constant factors.*

Theorem 8 gives a minimax lower bound for listwise ranking, which is the same as pairwise ranking. The proof is given in supplementary material. In [5], the authors have shown that for top-$k$ item selection under the MNL model, listwise comparisons can reduce the number of comparisons needed compared with pairwise comparisons. However, for exact ranking, listwise comparisons cannot.

# 4 Algorithm and the upper bound for pairwise ranking

In this section, we establish a (nearly) sample-complexity optimal $\delta$-correct algorithm for exact ranking, where whether the word "nearly" can be deleted depends on the structures of the instances. The algorithm is based on Binary Search proposed in [14] with upper bound $O(n\Delta_{\min}^{-2}\log(n/\delta))$, where $\Delta_{\min} := \min_{i \neq j} \Delta_{i,j}$. Binary Search has two limitations: (i) it requires the knowledge of $\Delta_{\min}$ *a priori* to run, and (ii) it does not utilize the unequal noise levels.

In this paper, we propose a technique named *Attempting with error prevention* and establish a corresponding insertion subroutine that attempts to insert an item $i$ into a sorted list with a guessing $\Delta_i$-value, while preventing errors from happening if the guess is not well chosen. If the guess is small enough, this subroutine correctly inserts the item with a large probability, and if not, this subroutine will, with a large probability, not insert the item into a wrong position. By attempting to insert item $i$ with diminishing guesses of $\Delta_i$, this subroutine finally correctly inserts item $i$ with a large confidence.

To implement the technique "Attempting with error prevention", we first need to construct a useful subroutine called Attempting-Comparison (ATC), which attempts to rank two items with $\epsilon$, a guess of $\Delta_{i,j}$. Then, by ATC, we establish Attempting-Insertion (ATI), which also adopts this technique.

---

**Subroutine 1** Attempting-Comparison$(i, j, \epsilon, \delta)$ (ATC)

---

**Initialize:** $\forall t$, let $b^t = \sqrt{\frac{1}{2t}\log\frac{\pi^2 t^2}{3\delta}}$; $b^{max} \leftarrow \lceil \frac{1}{2\epsilon^2}\log\frac{2}{\delta} \rceil$; $w_i \leftarrow 0$;

1: **for** $t \leftarrow 1$ to $b^{max}$ **do**
2:     Compare $i$ and $j$ once; Update $w_i \leftarrow w_i + 1$ if $i$ wins; Update $\hat{p}_i^t \leftarrow w_i/t$;
3:         **if** $\hat{p}_i^t > 1/2 + b^t$ **then return** $i$;
4:         **if** $\hat{p}_i^t < 1/2 - b^t$ **then return** $j$;
5: **end for**
6: **return** $i$ if $\hat{p}_i^t > 1/2$; **return** $j$ if $\hat{p}_i^t < 1/2$; and **return** a random item **if** $\hat{p}_i^t = 1/2$;

---

**Lemma 9** (Theoretical Performance of ATC)**.** *ATC terminates after at most $b^{max} = O(\epsilon^{-2}\log(1/\delta))$ comparisons and returns the more preferred item with probability at least $1/2$. Further, if $\epsilon \leq \Delta_{i,j}$, then ATC returns the more preferred item with probability at least $1 - \delta$.*

Next, to establish insertion subroutine ATI, we introduce preference interval trees [14] (PIT). A PIT is constructed from a sorted list of items. For a sorted list of items $S$ with size $l$, without loss of generality, we assume that $r_1 \succ r_2 \succ \cdots \succ r_l$. We introduce two artificial items $-\infty$ and $+\infty$, where $-\infty$ is such that $p_{i,-\infty} = 1$ for any item $i$, and $+\infty$ is such that $p_{i,+\infty} = 0$ for any item $i$.

**Preference Interval Tree** [14]. A preference interval tree constructed from the sorted list $S$ satisfies the following conditions: (i) It is a binary tree with depth $\lceil 1 + \log_2(|S| + 1) \rceil$. (ii) Each node $u$ holds an interval $(u.\text{left}, u.\text{right})$ where $u.\text{left}, u.\text{right} \in S \cup \{-\infty, +\infty\}$, and if $u$ is non-leaf, it holds an item $u.\text{mid}$ satisfying $u.\text{right} \succ u.\text{mid} \succ u.\text{left}$. (iii) A node $i$ is in the interval $(j, k)$ if and only if $k \succ i \succ j$. (iv) The root node is with interval $(-\infty, +\infty)$. From left to right, the leaf nodes are with intervals $(-\infty, r_l), (r_l, r_{l-1}), (r_{l-1}, r_{l-2}), ..., (r_2, r_1), (r_1, +\infty)$. (v) Each non-leaf node $u$ has two children $u.\text{lchild}$ and $u.\text{rchild}$ such that $u.\text{left} = u.\text{lchild}.\text{left}$, $u.\text{right} = u.\text{rchild}.\text{right}$ and $u.\text{mid} = u.\text{lchild}.\text{right} = u.\text{rchild}.\text{left}$.

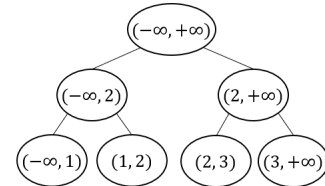

Figure 1: An example of PIT, constructed from a sorted list with three items $3 \succ 2 \succ 1$.

Based on the notion of PIT, we present insertion subroutine ATI in Subroutine 2. ATI runs a random walk on the PIT to insert $i$ into $S$. Let $X$ be the point that moves on the tree. We say a leaf $u_0$ *correct*

if the item $i$ belongs to $(u_0.\text{left}, u_0.\text{right})$. Define $d(X) :=$ the distance (i.e., the number of edges) between $X$ and $u_0$. At each round of the subroutine, if all comparisons give correct results, we say this round is *correct*, otherwise we say *incorrect*. For each correct round, either $d(X)$ is decreased by 1 or the counter of $u_0$ is increased by 1. The subroutine inserts $i$ into $u_0$ if $u_0$ is counted for $1 + \frac{5}{16}t^{\max}$ times. Thus, after $t^{\max}$ rounds, the subroutine correctly inserts $i$ into $S$ if the number of correct rounds is no less than $\frac{21}{32}t^{\max} + \frac{h}{2}$, where $h = \lceil 1 + \log_2(|S| + 1) \rceil$ is the depth of the tree. If guessing $\epsilon \le \Delta_i$, then each round is correct with probability at least $q$, making the subroutine correctly insert item $i$ with probability at least $1 - \delta$.

For all $\epsilon > 0$, each round is incorrect with probability at most $1/2$, and thus, by concentration inequalities, we can also show that with probability at least $1 - \delta$, $i$ will not be placed into any leaf node other than $u_0$. That is, if $\epsilon > \Delta_i$, the subroutine either correctly inserts $i$ or returns *unsure* with probability at least $1 - \delta$. The choice of parameters guarantees the sample complexity. Lemma 10 states its theoretical performance, and the proof is relegated to the supplementary material.

---

**Subroutine 2** Attempting-Insertion$(i, S, \epsilon, \delta)$ (ATI).

---

**Initialize:** Let $T$ be a PIT constructed from $S$;
$h \leftarrow \lceil 1 + \log_2(1 + |S|) \rceil$, the depth of $T$;
For all leaf nodes $u$ of $T$, initialize $c_u \leftarrow 0$;
Set $t^{\max} \leftarrow \lceil \max\{4h, \frac{512}{25} \log \frac{2}{\delta}\} \rceil$ and $q \leftarrow \frac{15}{16}$;

1: $X \leftarrow$ the root node of $T$;
2: **for** $t \leftarrow 1$ to $t^{\max}$ **do**
3:     **if** $X$ is the root node **then**
4:         **if** ATC$(i, X.\text{mid}, \epsilon, 1 - q) = i$ **then** $X \leftarrow X.\text{right};$     #i.e., ATC returns $i \succ X.\text{mid}$
5:         **else** $X \leftarrow X.\text{left};$
6:     **else if** $X$ is a leaf node **then**
7:         **if** ATC$(i, X.\text{left}, \epsilon, 1 - \sqrt{q}) = i \wedge$ ATC$(i, X.\text{right}, \epsilon, 1 - \sqrt{q}) = X.\text{right}$ **then**
8:             $c_X \leftarrow c_X + 1;$
9:             **if** $c_X > b^t := \frac{1}{2}t + \sqrt{\frac{t}{2} \log \frac{\pi^2 t^2}{3\delta}} + 1$ **then**
10:                 Insert $i$ into the corresponding interval of $X$ and **return** *inserted*;
11:         **else if** $c_X > 0$ **then** $c_X \leftarrow c_X - 1$
12:         **else** $X \leftarrow X.\text{parent}$
13:     **else**
14:         **if** ATC$(i, X.\text{left}, \epsilon, 1 - \sqrt[3]{q}) = X.\text{left} \vee$ ATC$(i, X.\text{right}, \epsilon, 1 - \sqrt[3]{q}) = i$ **then**
15:             $X \leftarrow X.\text{parent};$
16:         **else if** ATC$(i, X.\text{mid}, \epsilon, 1 - \sqrt[3]{q}) = i$ **then** $X \leftarrow X.\text{rchild};$
17:         **else** $X \leftarrow X.\text{lchild};$
18: **end for**
19: **if** there is a leaf node $u$ with $c_u \ge 1 + \frac{5}{16}t^{\max}$ **then**
20:     Insert $i$ into the corresponding interval of $u$ and **return** *inserted*;
21: **else return** *unsure*;

---

**Lemma 10** (Theoretical performance of ATI). *Let $\delta \in (0, 1)$. ATI returns after $O(\epsilon^{-2} \log(|S|/\delta))$ comparisons and, with probability at least $1 - \delta$, correctly inserts $i$ or returns unsure. Further, if $\epsilon \le \Delta_i$, it correctly inserts $i$ with probability at least $1 - \delta$.*

By Lemma 10, we can see that the idea "Attempting with error prevention" is successfully implemented. Thus, by repeatedly attempting to insert an item with diminishing guess $\epsilon$ with proper confidences for the attempts, one can finally correctly insert $i$ with probability $1 - \delta$. We use this idea to establish the insertion subroutine Iterative-Attempting-Insertion (IAI, Subroutine 3), and then use it to establish the ranking algorithm Iterative-Insertion-Ranking (IIR, Algorithm 4). Their theoretical performances are stated in Lemma 11 and Theorem 12, respectively, and their proofs are given in supplementary material.

**Lemma 11** (Theoretical Performance of IAI). *With probability at least $1 - \delta$, IAI correctly inserts $i$ into $S$, and conducts at most $O(\Delta_i^{-2}(\log \log \Delta_i^{-1} + \log(|S|/\delta)))$ comparisons.*

**Theorem 12** (Theoretical Performance of IIR). *With probability at least $1 - \delta$, IIR returns the exact ranking of $[n]$ and conducts at most $O(\sum_{i \in [n]} \Delta_i^{-2}(\log \log \Delta_i^{-1} + \log(n/\delta)))$ comparisons.*

| **Subroutine 3** Iterative-Attempting-Insertion (IAI). | **Algorithm 4** Iterative-Insertion-Ranking (IIR). |
|---|---|
| **Input parameters:** $(i, S, \delta)$; | **Input:** $S = [n]$, and confidence $\delta > 0$; |
| **Initialize:** For all $\tau \in \mathbb{Z}^+$, set $\epsilon_\tau = 2^{-(\tau+1)}$ and | 1: $Ans \leftarrow$ the list containing only $S[1]$; |
| $\delta_\tau = \frac{6\delta}{\pi^2 \tau^2}$; $t \leftarrow 0$; $Flag \leftarrow$ *unsure*; | 2: **for** $t \leftarrow 2$ to $|S|$ **do** |
|   1: **repeat** $t \leftarrow t + 1$; | 3:      IAI$(S[t], Ans, \delta/(n-1))$; |
|   2:      $Flag \leftarrow$ ATI$(i, S, \epsilon_t, \delta_t)$; | 4: **end for** |
|   3: **until** $Flag =$ *inserted* | 5: **return** $Ans$; |

**Remark:** We can see that the upper bounds of IIR depend on the values of $(\Delta_i, i \in [n])$ while the lower bounds given in Theorem 2 depend on the values of $(\tilde{\Delta}_i, i \in [n])$. Without SST, it is possible $\tilde{\Delta}_i < \Delta_i$, but if SST holds, then our algorithm is optimal up to a constant factor given $\delta \preceq 1/poly(n)$, or $\max_{i,j \in [n]} \tilde{\Delta}_i / \tilde{\Delta}_j \preceq O(n^{1/2-p})$ for some constant $p > 0$. According to [10, 11, 12], ranking without the SST condition can be much harder than that with SST , and it remains an open problem whether our upper bound is tight or not when the SST condition does not hold.

# 5 Numerical results

In this section, we provide numerical results to demonstrate the efficacy of our proposed IIR algorithm. We compare IIR with: (i) Active-Ranking (AR) [15], which focuses on the Borda-Score model and is not directly comparable to our algorithm. We use it as an example to show that although Borda-Ranking may be the same as exact ranking, for finding the exact ranking, the performance of Borda-Score algorithms is not always as good as that for finding the Borda-Ranking [7]; (ii) PLPAC-AMPR [35], an algorithm for PAC ranking under the MNL model. By setting the parameter $\epsilon = 0$, it can find the exact ranking with $O((n \log n) \max_{i \in [n]} \Delta_i^{-2} \log(n \Delta_i^{-1} \delta^{-1}))$ comparisons, higher than our algorithm by at least a log factor; (iii) UCB + Binary Search of [14]. In the Binary Search algorithm of [14], a subroutine that ranks two items with a constant confidence is required. In [14], it assumes the value of $\Delta_{\min} = \min_{i \in [n]} \Delta_i$ is priorly known, and the subroutine is simply comparing two items for $\Theta(\Delta_{\min}^{-2})$ times and returns the item that wins more. In this paper, the value of $\Delta_{\min}$ is not priorly known, and here, we use UCB algorithms such as LUCB [23] to play the role of the required subroutine. The UCB algorithms that we use include Hoeffding-LUCB [17, 23], KL-LUCB [2, 23], and lil'UCB [18]. For Hoeffding-LUCB and KL-LUCB, we choose $\gamma = 2$. For lil'UCB, we choose $\epsilon = 0.01$, $\beta = 1$, and $\lambda = (\frac{2+\beta}{\beta})^2$ [8].

**Instances.** The experiments are conducted on three different types of instances. To simplify notation, we use $r_1 \succ r_2 \succ \cdots \succ r_n$ to denote the true ranking, and let $\Delta = 0.1$. (i) Type-Homo: For any $r_i \succ r_j$, $p_{r_i, r_j} = 1/2 + \Delta$. (ii) Type-MNL: The preference score of $r_i$ (i.e., $\theta_{r_i}$) is generated by taking an independent instance of Uniform($[0.9 * 1.5^{n-i}, 1.1 * 1.5^{n-i}]$). By this, for any $i$, $\Delta_i$ is around 0.1. (iii) Type-Random: For any $r_i \succ r_j$, $p_{r_i, r_j}$ is generated by taking an independent instance of Uniform($[0.5 + 0.8\Delta, 0.5 + 1.5\Delta]$). By this, for any $i$, $\Delta_i$ is around 0.1. We let $\Delta_i$'s be close to 0.1 in order to decrease the influence of $\Delta_i$'s on sample complexities and show how the sample complexities of the algorithms grow with $n$.

The numerical results for these three types are presented in Figure 2 (a)-(c), respectively. For all simulations, we input $\delta = 0.01$. Every point of every figure is averaged over 100 independent trials. In every figure, for the same $n$-value, the algorithms are tested on an identical input instance.

From Figure 2, we can see that our algorithm significantly outperforms the existing algorithms. We can also see that the sample complexity of IIR scales with $n \log n$, which is consistent with our theoretical results. There are some insights about the practical performance of IIR. First, in Lines 3 and 4 of ATC and Lines 9 and 10 of ATI, we use LUCB-like [23] designs to allow the algorithms return before completing all required iterations, which does not improve the theoretical upper bound but can improve the practical performance. Second, in the theoretical analysis, we only show that

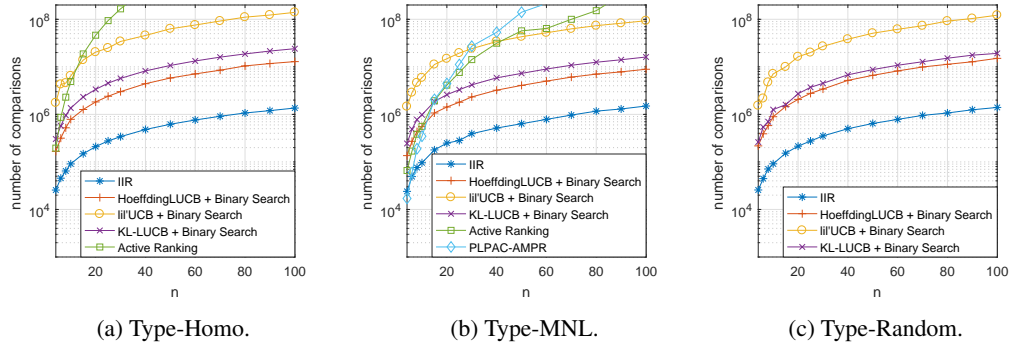

<div align="center">

(a) Type-Homo.       (b) Type-MNL.       (c) Type-Random.

Figure 2: Comparisons between IIR and existing methods.

</div>

ATI correctly inserts an item $i$ with high probability when inputting $\epsilon \leq \Delta_i$, but the algorithm may return before $\epsilon$ being that small, making the practical performance better than what the theoretical upper bound suggests.

## 6 Conclusion

In this paper, we investigated the theoretical limits of exact ranking with minimal assumptions. We do not assume any prior knowledge of the comparison probabilities and gaps, and derived the lower bounds and upper bound for instances with unequal noise levels. We also derived the model-specific pairwise and listwise lower bound for the MNL model, which further shows that in the worst case, listwise ranking is no more efficient than pairwise ranking in terms of sample complexity. The iterative-insertion-ranking (IIR) algorithm proposed in this paper indicates that our lower bounds are optimal under strong stochastic transitivity (SST) and some mild conditions. Numerical results also suggest that our ranking algorithm outperforms existing works in the literature.

**Acknowledgments**

This work has been supported in part by NSF grants ECCS-1818791, CCF-1758736, CNS-1758757, CNS-1446582, CNS-1901057; ONR grant N00014-17-1-2417; AFRL grant FA8750-18-1-0107, and by Institute for Information & communications Technology Promotion (IITP) grant funded by the Korea government (MSIT), (2017-0-00692, Transport-aware Streaming Technique Enabling Ultra Low-Latency AR/VR Services).

## Footnotes

[1] For any positive integer $k$, define $[k] := \{1, 2, ..., k\}$ to simplify notation

[2] All log in this paper, unless explicitly noted, are natural log.

[3]Stochastic triangle inequality means that for all items $i, j, k$ with $i \succ j \succ k$, $\Delta_{i,k} \leq \Delta_{i,j} + \Delta_{j,k}$.

[4]$poly(n)$ means a polynomial function of $n$, and $\delta \preceq 1/poly(n)$ means $\delta \preceq n^{-p}$ for some constant $p > 0$.

[5]For example, under a model with $\Delta_{i,j} = \Delta$ for any $i \neq j$ where $\Delta > 0$ is unknown, one may first estimate a lower bound of $\Delta$, and then perform algorithms in [14], yielding a sample complexity lower than Theorem 2.

[6]In this paper, "w.h.p." means with probability at least $1 - n^{-p}$, where $p > 0$ is a sufficiently large constant.

[7] For instance, when $p_{r_i, r_j} = 1/2 + \Delta$ for all $i < j$, the Borda-Score of item $r_i$ is $\frac{1}{n-1} \sum_{j \neq i} p_{r_i, r_j} = 1/2 + \frac{n+1-2i}{n-1}\Delta$, and $\Delta_{r_i} = \Theta(1/n)$. Thus, by [15], the sample complexity of AR is at least $O(n^3 \log n)$.

[8] We do not choose the combination ($\epsilon = 0$, $\beta = 1$, and $\lambda = 1 + 10/n$) that has a better practical performance because this combination does not have theoretical guarantee, making the comparison in some sense unfair.

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
