[Supplementary Material]

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

# Supplementary material

## A  Further discussions

### A.1  Non-$\delta$-correct algorithms

In Section 1, we define the notion of $\delta$-correct algorithms, which return correct results with probability at least $1 - \delta$ for any input instances satisfying assumptions A1 to A3 (defined in Section 1). It is reasonable to consider $\delta$-correct algorithms since we may not want an algorithm that performs pretty well on some instances but badly on others. However, to give better insights about $\delta$-correct algorithms and the lower bounds in Theorem 2, we give an algorithm that is not $\delta$-correct and has sample complexity lower than Theorem 2 for a specific class of instances.

**Example 13** (A non-$\delta$-correct algorithm). $\mathcal{A}$ *is an algorithm for ranking* 3 *items. It views each pair of items as a coin, and calls KL-LUCB [25] to find the pair* $(i, j)$ *with the largest* $p_{i,j}$-*value. Then, it claims that* $i$ *is the most preferred item and* $j$ *is the worst. Obviously,* $\mathcal{A}$ *is not* $\delta$-*correct for ranking* 3 *items. However, for an instance with* $p_{r_1,r_2} = 1/2 + \Delta$, $p_{r_1,r_3} = 1 - \Delta$, *and* $p_{r_2,r_3} = 1 - 2\Delta$, *where* $r_1 \succ r_2 \succ r_3$ *is the unknown true ranking and* $\Delta \in (0, 1/6)$ *is unknown, with probability at least* $1 - \delta$, *algorithm* $\mathcal{A}$ *finds its true ranking by using* $O(\Delta^{-1} \log(\Delta^{-1}\delta^{-1}))$ *comparisons.*

To see this upper bound, we first define some notations. For $p, q \in [0, 1]$, the KL-Divergence [9] between them is defined as $d(q, p) := D_{KL}(q||p) = q \log \frac{q}{p} + (1 - q) \log \frac{1-q}{1-p}$. The Chernoff-Information [25] between them is defined as $d^*(q, p) := d(z^*, p) = d(z^*, q)$, where $z^*$ is the unique $z$ such that $d(z, p) = d(z, q)$. According to [25, Theorem 3], the algorithm KL-LUCB distinguishes two coins (Bernoulli arms) with mean rewards $\lambda$ and $\mu$ by taking $O(\frac{1}{d^*(\lambda,\mu)} \log \frac{1}{\delta d^*(\lambda,\mu)})$ samples. In this instance, we observe that for a constant $c > 1$, $d(c\Delta, \Delta) = \Theta(\Delta)$. Thus, we have $d^*(1 - 2\Delta, 1 - \Delta) = d^*(2\Delta, \Delta) = \Theta(\Delta)$. Hence, KL-LUCB distinguishes $p_{r_1,r_3}$ and $p_{r_2,r_3}$ by $O(\Delta^{-1} \log(\delta^{-1}\Delta^{-1}))$ comparisons. Since the gap between $p_{r_1,r_2}$ and $p_{r_1,r_3}$ is even larger, they can also be distinguished by the above number of comparisons. This shows the upper bound, which suggests that the $\Delta_i^{-2}$ term is not necessary for non-$\delta$-correct algorithms.

We note that $\mathcal{A}$ does not need any information of this instance a priori to run. Although it is not $\delta$-correct, it can solve this class of instances with sample complexity lower than Theorem 2. However, in general, this algorithm may be of no sense as it only works for a restricted class of instances. This is the reason why we want to bound the sample complexity of $\delta$-correct algorithms but not that of arbitrary ones, as there may always exist non-$\delta$-correct algorithms that have extremely good performance on some restricted class of instances.

### A.2  An instance where Eq. (2) does not hold as a lower bound

When $\delta$ is a positive constant and $\max_{i,j\in[n]} \tilde{\Delta}_i / \tilde{\Delta}_j \succ \sqrt{n}$, the lower bound given in Eq. (2) may not hold. In this subsection, we give an example such that Eq. (2 does not hold as a lower bound.

**Example 14** (An example that Eq. (2) does not hold as a lower bound). *Assume that* $r_1 \succ r_2 \succ \cdots \succ r_n$ *is the unknown true ranking. Suppose* $\delta = 1/4$, $\Delta_{r_1,r_2} = n^{-10}$ *and* $\Delta_{r_i,r_j} = 0.01$ *for all* $\{r_i, r_j\} \neq \{r_1, r_2\}$. *For this instance, there is a* $(1/4)$-*correct algorithm that finds its true ranking with confidence* $3/4$ *by* $O(n^{20} \log \log n + n^2 \log n)$ *comparisons, which is lower than Eq. (2):* $\Omega(n^{20} \log n + n \log n)$. *This implies that Eq. (2) does not hold as a lower bound in this case.*

To see the upper bound, we can view each pair as a coin (aka Bernoulli arms), and then use lil'UCB [20] to find the pair with the least gap (i.e., $\Delta_{i,j}$) with confidence $11/12$. According to [20], this step takes $O(n)$ comparisons. Then, we rank the pair with the smallest gap with $11/12$ confidence. This step takes $O(\Delta_{r_1,r_2}^{-2} \log \log \Delta_{r_1,r_2}^{-1}) = O(n^{20} \log \log n)$ comparisons. Finally, we rank all other pairs with $1 - \frac{1}{12n^2}$ confidence for each, and this step takes $O(n^2 \log n)$ comparisons. After ranking all pairs of items, the true ranking is found, and thus, the total sample complexity is $O(n^{20} \log \log n + n^2 \log n)$.

For this instance, the lower bound in Eq. (2) is $\Omega(n^{20} \log n + n \log n)$, higher than the upper bound. Thus, when the given condition does not hold, the lower bound in Eq. (2) may not hold. However, there is at most a log gap, and the lower bound in Eq. (1) does not need this condition.

## B  Proofs

### B.1  Proof of Theorem 2

**Theorem 2** (Lower bound for pairwise ranking). *Given $\delta \in (0, 1/12)$ and an instance $\mathcal{I}$ with $n$ items, then the number of comparisons used by a $\delta$-correct algorithm $\mathcal{A}$ with no prior knowledge about the gaps of $\mathcal{I}$ is lower bounded by*

$$\Omega\big( \sum_{i \in [n]} [\tilde{\Delta}_i^{-2} (\log \log \tilde{\Delta}_i^{-1} + \log(1/\delta))] + \min\{ \sum_{i \in [n]} \tilde{\Delta}_i^{-2} \log(1/x_i) : \sum_{i \in [n]} x_i \leq 1\} \big). \quad (1)$$

*If $\delta \preceq 1/poly(n)^9$, or $\max_{i,j \in [n]} \{\tilde{\Delta}_i / \tilde{\Delta}_j\} \preceq n^{1/2-p}$ for some constant $p > 0$, then the lower bound becomes*

$$\Omega\big( \sum_{i \in [n]} \tilde{\Delta}_i^{-2} (\log \log \tilde{\Delta}_i^{-1} + \log(n/\delta)) \big). \quad (2)$$

*Proof.* **Step 1** is to prove the lower bound for ranking two items, which is stated in Lemma 15. In the proof of Lemma 15, we will make use of the results in [15, 20, 27]. The proof can be found in Section B.12

**Lemma 15** (Lower bound for ranking two items). *Let $\delta \in (0, 1/4)$ and $\delta$-correct algorithm $\mathcal{A}_2$ be given. Let $T_{\mathcal{A}_2}(\Delta_{i,j})$ be the number of comparisons conducted by $\mathcal{A}_2$ under the $\Delta_{i,j}$-values. To rank $i$ and $j$ with error probability no more than $\delta$, there is a universal constant $c_{lb2} > 0$ such that*

$$\limsup_{\Delta_{i,j} \to 0} \frac{\mathbb{E}[T_{\mathcal{A}_2}(\Delta_{i,j})]}{\Delta_{i,j}^{-2} (\log \log \Delta_{i,j}^{-2} + \log \delta^{-1})} \geq c_{lb2}. \quad (3)$$

**Step 2** is to define problems $\mathcal{P}_1$ and $\mathcal{P}_2$. Let $(r_1, r_2, ..., r_n)$ be a given permutations of $[n]$ and assume that $q_1 \succ q_2 \succ \cdots \succ q_n$ is the unknown true ranking. Assume that $n$ is odd (when $n$ is even, we can prove the same results similarly), and say $n = 2m + 1$. A pair $(r_i, r_j)$ is said to be *significant* if there exists an $k$ in $[m]$ such that $\{r_i, r_j\} = \{r_{2k-1}, r_{2k}\}$, and *insignificant* otherwise.

Define a set $\Pi := \{0, 1\}^m$. For any $\vec{\pi} = (\pi_1, \pi_2, ..., \pi_m) \in \Pi$, define a corresponding hypothesis $\mathcal{H}_{\vec{\pi}}$ that claims: (i) the true ranking of $[n]$ is $s_1 \succ s_2 \succ \cdots \succ s_n$; (ii) $s_n = r_n$; (iii) for any $k \in [m]$, $(s_{2k-1}, s_{2k}) = (r_{2k-1}, r_{2k})$ if $\pi_k = 1$, and $(s_{2k-1}, s_{2k}) = (r_{2k}, r_{2k-1})$ otherwise; (iv) for any insignificant pair $(r_i, r_j)$, the probability that $r_i$ wins a comparison over the pair $(r_i, r_j)$ is $p^{\vec{\pi}}_{r_i, r_j} = p_{r_i, r_j}$; (v) For any $k \in [m]$ and the corresponding significant pair $(r_{2k-1}, r_{2k})$, the probability that $r_{2k-1}$ wins a comparison over the pair $(r_{2k-1}, r_{2k})$ is $p^{\vec{\pi}}_{r_i, r_j} = 1/2 + \Delta_{r_{2k-1}, r_{2k}}$ if $\pi_k = 1$, and is $(1/2 - \Delta_{r_{2k-1}, r_{2k}})$ otherwise. In other words, $\mathcal{H}_{\vec{\pi}}$ claims a true ranking that is almost the same as $r_1 \succ r_2 \succ \cdots \succ r_n$ but the positions of $(r_{2k-1}, r_{2k})$ are interchanged for all $k \in [m]$ such that $\pi_k = 0$. E.g., for $n = 3$ and $\vec{\pi} = (0)$, $\mathcal{H}_{\vec{\pi}}$ claims that the true ranking is $r_2 \succ r_1 \succ r_3$, $p^{\vec{\pi}}_{r_1, r_2} = 1/2 - \Delta_{r_1, r_2}$, $p^{\vec{\pi}}_{r_1, r_3} = p_{r_1, r_3}$, and $p^{\vec{\pi}}_{r_2, r_3} = p_{r_2, r_3}$.

We further assume that there is a $\vec{\pi}_0 \in \Pi$ such that $\mathcal{H}_{\vec{\pi}_0}$ is true, and each $\vec{\pi} \in \Pi$ has the same prior probability to be $\vec{\pi}_0$.

**Problem $\mathcal{P}_1$.** Knowing the fact that there exists a $\pi^0 \in \Pi$ such that $\mathcal{H}_{\vec{\pi}^0}$ is true, we want to find $\pi^0$ with confidence $1 - \delta$, and use as few comparisons as possible.

Next, we start defining problem $\mathcal{P}_2$. An instance of $\mathcal{P}_2$ involves $\binom{n}{2}$ coins, and each is indexed by an element of $\{(i, j) : i, j \in [n] \wedge i < j\}$. We use $C_{i,j}$ to denote the coin indexed by $(i, j)$. For each coin $C_{i,j}$, each toss of it gives a head with probability $\mu_{i,j}$, and gives a tail with probability $1 - \mu_{i,j}$. We name $\mu_{i,j}$ as the *head probability* of coin $C_{i,j}$. We assume that the outcomes of tosses are independent across coins and time. Similar to the items, coin $C_{i,j}$ is said to be *significant* if there is a $k$ such that $(i, j) = (2k - 1, 2k)$, and is *insignificant* otherwise. We assume that for all insignificant coins $C_{i,j}$, $\mu_{i,j} = p_{r_i, r_j}$, and for all significant coins $C_{2k-1, 2k}$, $\mu_{2k-1,2k} = 1/2 + \Delta_{r_{2k-1}, r_{2k}}$ or $1/2 - \Delta_{r_{2k-1}, r_{2k}}$, either has a prior probability $1/2$ to be true.

**Problem $\mathcal{P}_2$.** With probability $\geq 1 - \delta$, we want to find whether $\mu_{2k-1, 2k} > 1/2$ for all $k \in [m]$.

**Step 3** is to show the following lemma, which states that $\mathcal{P}_2$ can be reduced to $\mathcal{P}_1$, and $\mathcal{P}_1$ can be reduced to exact ranking. Its proof can be found in Section B.13.

**Lemma 16** (Reductions). *With the above definitions, (i) if the true ranking of $[n]$ is found, with no more comparisons, one can get the solution of $\mathcal{P}_1$, and (ii) if an algorithm solves $\mathcal{P}_1$ with $N$ expected number of comparison, there is another algorithm that solves $\mathcal{P}_2$ with $N$ expected number of tosses.*

**Step 4** is to prove the following lemma regarding the lower bound of problem $\mathcal{P}_2$. Its proof can be found in Section B.14

**Lemma 17.** *For $\delta \in (0, 1/12)$, the expected number of tosses needed for solving $\mathcal{P}_2$ is at least*

$$\Omega\Big( \sum_{k\in[m]} \Delta_{q_{2k-1},q_{2k}}^{-2} \cdot \log\log \Delta_{q_{2k-1},q_{2k}}^{-1} + \min\{ \sum_{k\in[m]} \Delta_{q_{2k-1},q_{2k}}^{-2} \cdot \log(\delta_k^{-1}) : \sum_{k\in[m]} \delta_k \le 2\delta \}\Big). \quad (4)$$

**Step 5** is to prove the lower bound given in Eq. (1). Lemmas 16 proves that we can reduce $\mathcal{P}_2$ to $\mathcal{P}_1$ and reduce $\mathcal{P}_1$ to exact ranking. Lemma 17 states a lower bound on $\mathcal{P}_2$. Thus, by Lemmas 16 and 17, we have that the sample complexity of exact ranking is lower bounded by (4).

We can construct a similar problem to $\mathcal{P}_2$, and by the similar steps as in the proof of Lemma 17, we have that the sample complexity of exact ranking is also lower bounded by

$$\Omega\Big( \sum_{k\in[m]} \Delta_{q_{2k},q_{2k+1}}^{-2} \log\log \Delta_{q_{2k},q_{2k+1}}^{-1} + \min\{ \sum_{k\in[m]} \Delta_{q_{2k},q_{2k+1}}^{-2} \log(1/\delta_k) : \sum_{k\in[m]} \delta_k \le 2\delta \}\Big). \quad (5)$$

We recall that $q_1 \succ q_2 \succ \cdots \succ q_n$ is the true ranking. Since for any $i \in [n]$, $\tilde{\Delta}_{q_i} = \Delta_{q_i,q_{i-1}} \wedge \Delta_{q_i,q_{i+1}}$, we have

$$\begin{aligned}
\mathbb{E}N_{\mathcal{A}}(\mathcal{I}) &\succeq \sum_{k\in[m]} \Delta_{q_{2k-1},q_{2k}}^{-2} \log\log \Delta_{q_{2k-1},q_{2k}}^{-1} + \sum_{k\in[m]} \Delta_{q_{2k},q_{2k+1}}^{-2} \log\log \Delta_{q_{2k},q_{2k+1}}^{-1} \\
&\ge \sum_{k\in[m]} \max\{\Delta_{q_{2k-1},q_{2k}}^{-2} \log\log \Delta_{q_{2k-1},q_{2k}}^{-1}, \Delta_{q_{2k},q_{2k+1}}^{-2} \log\log \Delta_{q_{2k},q_{2k+1}}^{-1}\} \\
&= \sum_{k\in[m]} \tilde{\Delta}_{q_{2k}}^{-2} \log\log \tilde{\Delta}_{q_{2k}}^{-1} \\
&\overset{(a)}{\ge} \frac{1}{3} \sum_{i=1}^{n} \tilde{\Delta}_i^{-2} \log\log \tilde{\Delta}_i^{-1}, \quad (6)
\end{aligned}$$

where (a) holds because for any $k \in [m]$, $\tilde{\Delta}_{q_{2k+1}} = \Delta_{q_{2k},q_{2k+1}} \wedge \Delta_{q_{2k+1},q_{2k+2}} \ge \tilde{\Delta}_{q_{2k}} \wedge \tilde{\Delta}_{q_{2k+2}}$.

We also have

$$\begin{aligned}
&\min\{ \sum_{k\in[m]} \Delta_{q_{2k-1},q_{2k}}^{-2} \log\delta_k^{-1} : \sum_{k\in[m]} \delta_k \le 2\delta \} + \min\{ \sum_{k\in[m]} \Delta_{q_{2k},q_{2k+1}}^{-2} \log\delta_k^{-1} : \sum_{k\in[m]} \delta_k \le 2\delta \} \\
&= \min\{ \sum_{k\in[m]} [\Delta_{q_{2k-1},q_{2k}}^{-2} \log(1/\delta_k) + \Delta_{q_{2k},q_{2k+1}}^{-2} \log(1/\delta_k')] : \sum_{k\in[m]} \delta_k \le 2\delta, \sum_{k\in[m]} \delta_k' \le 2\delta \} \\
&\ge \min\{ \sum_{k\in[m]} [\Delta_{q_{2k-1},q_{2k}}^{-2} + \Delta_{q_{2k},q_{2k+1}}^{-2}] \log\frac{1}{\delta \vee \delta'} : \sum_{k\in[m]} \delta_k \le 2\delta, \sum_{k\in[m]} \delta_k' \le 2\delta \} \\
&\ge \min\{ \sum_{k\in[m]} \tilde{\Delta}_{q_{2k}}^{-2} \log\frac{1}{\delta_k \vee \delta_k'} : \sum_{k\in[m]} \delta_k \le 2\delta, \sum_{k\in[m]} \delta_k' \le 2\delta \} \\
&\ge \min\{ \sum_{k\in[m]} \tilde{\Delta}_{q_{2k}}^{-2} \log\frac{1}{\delta_k \vee \delta_k'} : \sum_{k\in[m]} \delta_k \vee \delta_k' \le 4\delta \} \\
&\ge \min\{ \sum_{k\in[m]} \tilde{\Delta}_{q_{2k}}^{-2} \log(1/\delta_k) : \sum_{k\in[m]} \delta_k \le 4\delta \} \\
&\ge \min\{ \frac{1}{3} \sum_{i\in[n]} \tilde{\Delta}_i^{-2} \log(1/x_i) : \sum_{i\in[n]} x_i \le 12\delta \}. \quad (7)
\end{aligned}$$

By (7), first, we obtain that, for all $\delta \in (1, 1/12)$,

$$\mathbb{E}N_{\mathcal{A}}(\mathcal{I}) \succeq \sum_{i \in [n]} \tilde{\Delta}_i^{-2} \log(1/\delta). \tag{8}$$

Also, since $\delta < 1/12$, we obtain the lower bound

$$\mathbb{E}N_{\mathcal{A}}(\mathcal{I}) \succeq \min\{\sum_{i \in [n]} \tilde{\Delta}_i^{-2} \log(1/x_i) : \sum_{i \in [n]} x_i \leq 12\delta\}$$

$$\geq \min\{\sum_{i \in [n]} \tilde{\Delta}_i^{-2} \log(1/x_i) : \sum_{i \in [n]} x_i \leq 1\}. \tag{9}$$

The lower bound in Eq. (1) follows from summing up Equations (6), (8), and (9). This prove the lower bound in Eq. (2).

**Step 6** is to deduce the lower bound in Eq. (2) from Eq. (1).

**Case 1.** We consider the cases where $\delta \preceq 1/poly(n)$. We observe that, when $\delta \preceq 1/poly(n)$, $\log(1/\delta) \succeq \log n$. Thus, in Eq. (9), setting all $x_i = 1/n$, we have

$$\min\{\sum_{i \in [n]} \tilde{\Delta}_i^{-2} \log(1/x_i) : \sum_{i \in [n]} x_i \leq 1\} \leq \sum_{i \in [n]} \tilde{\Delta}_i^{-2} \log n \preceq \sum_{i \in [n]} \tilde{\Delta}_i^{-2} \log(1/\delta).$$

This means that the term $\min\{\cdots\}$ is dominated by the term $\sum_{i \in [n]} \tilde{\Delta}_i^{-2} \log(1/\delta)$. We also have $\sum_{i \in [n]} \tilde{\Delta}_i^{-2} \log(1/\delta) \simeq \sum_{i \in [n]} \tilde{\Delta}_i^{-2} \log(n/\delta)$ since $\log \delta^{-1} \succeq \log n$. Thus,

$$\sum_{i \in [n]} \tilde{\Delta}_i^{-2} \log(1/\delta) + \min\{\sum_{i \in [n]} \tilde{\Delta}_i^{-2} \log(1/x_i) : \sum_{i \in [n]} x_i \leq 1\} \simeq \sum_{i \in [n]} \tilde{\Delta}_i^{-2} \log(n/\delta),$$

which implies that when $\delta = 1/poly(n)$, the lower bound in (2) holds.

**Case 2.** We consider the case where $\max_{i,j \in [n]}\{\tilde{\Delta}_i/\tilde{\Delta}_j\} \leq c \cdot n^{1/2-p}$ for some constants $c, p > 0$. When this condition holds, for any $x_1, x_2, ..., x_n$ with $\sum_{i \in [n]} x_i \leq 1$, we have

$$\sum_{i \in [n]} \tilde{\Delta}_i^{-2} \log(1/x_i) = \sum_{j \in [n]} \tilde{\Delta}_j^{-2} \sum_{i \in [n]} \frac{\tilde{\Delta}_i^{-2}}{\sum_{j \in [n]} \tilde{\Delta}_j^{-2}} \cdot \log(1/x_i)$$

$$\overset{(a)}{\geq} \sum_{j \in [n]} \tilde{\Delta}_j^{-2} \cdot \log \frac{1}{\sum_{i \in [n]} x_i \cdot \frac{\tilde{\Delta}_i^{-2}}{\sum_{j \in [n]} \tilde{\Delta}_j^{-2}}}$$

$$\geq \sum_{j \in [n]} \tilde{\Delta}_j^{-2} \cdot \log \frac{1}{\sum_{i \in [n]} x_i \frac{1}{\sum_{j \in [n]} (c \cdot n^{-1/2+p})^2}}$$

$$\overset{(b)}{\geq} \sum_{j \in [n]} \tilde{\Delta}_j^{-2} \log \left[ \sum_{i \in [n]} (c \cdot n^{-1/2+p})^2 \right]$$

$$\geq \sum_{j \in [n]} \tilde{\Delta}_j^{-2} \log(c^2 n^{2p})$$

$$\succeq \sum_{i \in [n]} \tilde{\Delta}_i^{-2} \log n,$$

where (a) is due to the convexity of the functions $(\log(1/x_i), i \in [n])$, and (b) is due to $\sum_{k \in [n]} \delta_k \leq 1$. Thus, in this case,

$$\mathbb{E}N_{\mathcal{A}}(\mathcal{I}) \succeq \sum_{i \in [n]} \tilde{\Delta}_i^{-2} \log \log \Delta_i^{-1} + \sum_{i \in [n]} \tilde{\Delta}_i^{-2} \log n + \sum_{i \in [n]} \tilde{\Delta}_i^{-2} \log \delta^{-1}$$

$$= \sum_{i \in [n]} \tilde{\Delta}_i^{-2} \log \log \Delta_i^{-1} + \sum_{i \in [n]} \tilde{\Delta}_i^{-2} \log(n/\delta),$$

which is the lower bound in (2). This completes the proof of (2) and Theorem 2. $\qquad\square$

## B.2 Proof of Theorem 3

**Theorem 3.** *[Lower bound for the MNL model] Let $\delta \in (0, 1/12)$ and given a $\delta$-correct algorithm $\mathcal{A}$ with the knowledge that the input instances satisfy the MNL model, let $N_{\mathcal{A}}$ be the number of comparisons conducted by $\mathcal{A}$, then $\mathbb{E}[N_{\mathcal{A}}]$ is lower bounded by Eq. (1) with a different hidden constant factor. When $\delta \preceq 1/poly(n)$ or $\max_{i,j \in [n]}\{\Delta_i/\Delta_j\} \preceq n^{1/2-p}$ for some constant $p > 0$, the sample complexity is lower bounded by Eq. (2) with a different hidden constant factor.*

*Proof.* We prove this theorem by Lemmas 4, 5 and 6, which could be of independent interest. The proofs of these three lemmas can be found in Sections B.3, B.4, and B.5

Suppose that there are two coins with unknown *head probabilities* (the probability that a toss produces a head) $\lambda$ and $\mu$, respectively, and we want to find the more biased one (i.e., the one with the larger head probability). Lemma 4 states a lower bound on the number of heads or tails generated for finding the more biased coin, which works even if $\lambda$ and $\mu$ go to 0. This is in contrast to the lower bounds on the number of tosses given by previous works [20, 23, 27], which go to infinity as $\lambda$ and $\mu$ go to 0.

**Lemma 4** (Lower bound on number of heads)**.** *Let $\lambda + \mu \leq 1$, $\Delta := |\lambda/(\lambda + \mu) - 1/2|$, and $\delta \in (0, 1/2)$ be given. To find the more biased coin with probability $1 - \delta$, any $\delta$-correct algorithm for this problem produces $\Omega(\Delta^{-2}(\log \log \Delta^{-1} + \log \delta^{-1}))$ heads in expectation.*

Now we consider $n$ coins $C_1, C_2, ..., C_n$ with mean rewards $\mu_1, \mu_2, ..., \mu_n$, respectively, where for any $i \in [n]$, $\theta_i/\mu_i = c$ for some constant $c > 0$. Define the gaps of coins $\Delta_{i,j}^c := |\mu_i/(\mu_i + \mu_j) - 1/2|$, and $\Delta_i^c := \min_{j \neq i} \Delta_{i,j}^c$. We can check that for all $i$ and $j$, $\Delta_{i,j}^c = \Delta_{i,j}$, and $\Delta_i = \tilde{\Delta}_i = \Delta_i^c$.

**Lemma 5** (Lower bound for arranging coins)**.** *For $\delta < 1/12$, to arrange these coins in ascending order of head probabilities, the number of heads generated by any $\delta$-correct algorithm is lower bounded by Eq. (1) with a (possibly) different hidden constant factor.*

The next lemma shows that any algorithm solves a ranking problem under the MNL model can be transformed to solve the pure exploration multi-armed bandit (PEMAB) problem with Bernoulli rewards. Previous works [1, 17, 18] have shown that certain types of pairwise ranking problems (e.g., Borda-Score ranking) can also be transformed to PEMAB problems. But in this paper, we make a *reverse connection* that bridges these two classes of problems, which may be of independent interest.

**Lemma 6** (Reducing PEMAB problems to ranking)**.** *If there is a $\delta$-correct algorithm that correctly ranks $[n]$ with probability $1 - \delta$ by $M$ expected number of comparisons, then we can construct another $\delta$-correct algorithm that correctly arranges the coins $C_1, C_2, ..., C_n$ in the order of ascending head probabilities with probability $1 - \delta$ and produces $M$ heads in expectation.*

Combining Lemmas 5 and 6, we have that $\mathbb{E}[N_{\mathcal{A}}]$ is lower bounded by Eq. (1) with a different hidden constant factor. Then, by the same steps as the Step 6 of the proof of Theorem 2, we have that when $\delta \preceq 1/poly(n)$ or $\max_{i,j \in [n]}\{\Delta_i/\Delta_j\} \preceq n^{1/2-p}$ for some constant $p > 0$, $\mathbb{E}[N_{\mathcal{A}}]$ is lower bounded by Eq. (2) with a different hidden constant factor. This completes the proof. We omit the repetition for brevity and note that under the pairwise MNL model, $\Delta_i = \tilde{\Delta}$ for any item $i$, as the pairwise MNL model satisfies the SST condition. $\qquad\square$

## B.3 Proof of Lemma 4

**Lemma 4** (Lower bound on number of heads)**.** *Let $\lambda + \mu \leq 1$, $\Delta := |\lambda/(\lambda + \mu) - 1/2|$, and $\delta \in (0, 1/2)$ be given. To find the more biased coin with probability $1 - \delta$, any $\delta$-correct algorithm for this problem produces $\Omega(\Delta^{-2}(\log \log \Delta^{-1} + \log \delta^{-1}))$ heads in expectation.*

*Proof.* By contradiction, suppose that there is an algorithm $\mathcal{A}$ that does not satisfy the stated lower bound. We will show a contradiction to Lemma 15.

Given a coin with head probability $p = 1/2 + \eta$, where $\eta \in (-1/4, 0) \cup (0, 1/4)$ is unknown, we will use $\mathcal{A}$ to construct an algorithm to recover the value of $sign(\eta)$, i.e. the sign of $\eta$. Choose an $\alpha \in (0, 1)$. We recall that a $p$-coin denotes a coin such that each toss of it produces a head with probability $p$, and a tail otherwise.

Now, we construct two i.i.d. sequences of random variables: $\{X^t\}_{t=1}^{\infty}$ and $\{Y^t\}_{t=1}^{\infty}$.

Sequence $\{X^t\}_{t=1}^{\infty}$ is generated as follows: For any $t \in \mathbb{Z}^+$, with probability $\alpha$, we toss the $p$-coin, and assign $X^t = 1$ if the toss gives a head, and assign $X^t = 0$ otherwise. With probability $1 - \alpha$, we assign $X^t = 0$.

Sequence $\{Y^t\}_{t=1}^{\infty}$ is generated as follows: For any $t \in \mathbb{Z}^+$, with probability $\alpha$, we toss the $p$-coin, and assign $Y^t = 1$ if the toss gives a tail, and assign $Y^t = 0$ otherwise. With probability $1 - \alpha$, we assign $Y^t = 0$.

As a result, $(X^t, t \in \mathbb{Z}^+)$ are i.i.d. Bernoulli$(\lambda)$, and $(Y^t, t \in \mathbb{Z}^+)$ are i.i.d. Bernoulli$(\mu)$, respectively. Thus, we can view that $X^t$'s are generated by a $\lambda$-coin and $Y^t$'s are generated by a $\mu$-coin, where $\lambda = \alpha(1/2 + \eta)$ and $\mu = \alpha(1/2 - \eta)$. We check that $|\lambda/(\lambda + \mu) - 1/2| = \eta$.

Next, we use algorithm $\mathcal{A}$ to find the more biased one of $(X^t, t \in \mathbb{Z}^+)$ and $(Y^t, t \in \mathbb{Z}^+)$. If the result is $X^t$'s, then we decide $\eta > 0$, and if the result is $Y^t$'s, then we decide $\eta < 0$. According to the assumption, $\mathcal{A}$ finds the results with probability at least $1 - \eta$ and the number of times $t$ such that $X^t = 1$ or $Y^t = 1$ is at most $o(\eta^{-2}(\log \log \eta^{-1} + \log \delta^{-1}))$ in expectation. For each $t$ with $X^t = 1$ or $Y^t = 1$, the $p$-coin is tossed for at most 4 times in expectation (since $1/4 < p < 3/4$).

Thus, we can determine whether $\eta < 0$ or $\eta > 0$ (equivalent to ranking two items $i$ and $j$ with $p_{i,j} = 1/2 + \eta$) by $o(\eta^{-2}(\log \log \eta^{-1} + \log \delta^{-1}))$ tosses in expectation, contradicting Lemma 15. Thus, such an algorithm $\mathcal{A}$ does not exist. This completes the proof of Lemma 4. $\qquad\square$

### B.4 Proof of Lemma 5

**Lemma 5** (Lower bound for arranging coins)**.** *For $\delta < 1/12$, to arrange these coins in ascending order of head probabilities, the number of heads generated by any $\delta$-correct algorithm is lower bounded by Eq. (1) with a (possibly) different hidden constant factor.*

*Proof.* To prove this lemma, we need to show the following lower bound:

$$\Omega\Big( \sum_{i \in [n]} [\tilde{\Delta}_i^{-2}(\log \log \tilde{\Delta}_i^{-1} + \log(1/\delta))] + \min\{\sum_{i \in [n]} \tilde{\Delta}_i^{-2} \log(1/x_i) : \sum_{i \in [n]} x_i \leq 1\}\Big).$$

The proof is similar to that of Lemma 17. We assume that the true order of these coins is $(q_1, q_2, ..., q_n)$, and $n = 2m + 1$ is odd. When $n$ is even, we can prove the results in similar steps.

To arrange the coins in the ascending order of head probabilities, one at least needs to distinguish the orders of the pairs $(q_1, q_2), (q_3, q_4), ..., (q_{2m-1}, q_{2m})$. For any $k$ in $[m]$, to order $q_{2k-1}$ and $q_{2k}$ with probability $1 - \delta_k$, by Lemma 4, any $\delta$-correct algorithm generates $\Omega(\Delta_{q_{2k-1}, q_{2k}}^{-2}(\log \log \Delta_{q_{2k-1}, q_{2k}} + \log \delta_k^{-1}))$ heads in expectation. Thus, by the same steps as in the proof of Lemma 17, we obtain a lower bound as follows:

$$\Omega\Big( \sum_{k \in [m]} \Delta_{q_{2k-1}, q_{2k}}^{-2} \log \log \Delta_{q_{2k-1}, q_{2k}}^{-1} + \min\{\sum_{k \in [m]} \Delta_{q_{2k-1}, q_{2k}}^{-2} \log \delta_k^{-1} : \sum_{k \in [m]} \delta_k \leq 2\delta\}\Big).$$

Also, to find to orders of the pairs $(q_2, q_3), (q_4, q_5), ...(q_{2m}, q_{2m+1})$, there is another lower bound shown below:

$$\Omega\Big( \sum_{k \in [m]} \Delta_{q_{2k}, q_{2k+1}}^{-2} \log \log \Delta_{q_{2k}, q_{2k+1}}^{-1} + \min\{\sum_{k \in [m]} \Delta_{q_{2k}, q_{2k+1}}^{-2} \log \delta_k^{-1} : \sum_{k \in [m]} \delta_k \leq 2\delta\}\Big).$$

By the same steps as the Step 5 of the proof of Theorem 2, we can get the desired lower bound. We omit the repetition for brevity. This completes the proof. $\qquad\square$

### B.5 Proof of Lemma 6

**Lemma 6** (Reducing PEMAB problems to ranking)**.** *If there is a $\delta$-correct algorithm that correctly ranks $[n]$ with probability $1 - \delta$ by $M$ expected number of comparisons, then we can construct another $\delta$-correct algorithm that correctly arranges the coins $C_1, C_2, ..., C_n$ in the order of ascending head probabilities with probability $1 - \delta$ and produces $M$ heads in expectation.*

***Proof.*** To prove this lemma, consider the following procedure $\mathcal{A}_c$.

---

**Algorithm** Procedure $\mathcal{A}_c$

---
**Input:** Two coins $C_i$ and $C_j$ with unknown head probabilities $\mu_i$ and $\mu_j$, respectively;

  1: **repeat**
  2:     Randomly choose a coin $C_w$ and toss it;
  3:     Let $s \leftarrow 1$ if the the toss gives a head, and $s \leftarrow 0$ otherwise;
  4: **until** $s = 1$
  5: **return** $C_w$;

---

**Claim 18.** *Procedure $\mathcal{A}_c$ returns coin $C_i$ with probability $\mu_i/(\mu_i + \mu_j)$ and returns $C_j$ otherwise.*

***Proof of Claim 18.*** Let $T$ be the number of tosses conducted before $\mathcal{A}_c$ returns, and $X$ be the coin it returns. By using conditional probability, we have that for all $t \geq 1$ and $i$ in $[m]$,

$$\mathbb{P}\{T = t, X = C_i\} = \prod_{\tau=1}^{t-1} \mathbb{P}\{T > \tau \mid T > \tau - 1\} \cdot \mathbb{P}\{T = t, X = C_i \mid T > t - 1\}$$

$$= (\mathbb{P}\{T > 1\})^{t-1} \cdot \mathbb{P}\{T = 1, X = C_i\}$$

$$= \left(1 - \frac{1}{2}(\mu_i + \mu_j)\right)^{t-1} \cdot \frac{1}{2}\mu_i,$$

$$\mathbb{P}\{X = C_i\} = \sum_{t=1}^{\infty} \mathbb{P}\{T = t, X = C_i\}$$

$$= \sum_{t=1}^{\infty} \left(1 - \frac{1}{2}(\mu_i + \mu_j)\right)^{t-1} \cdot \frac{1}{2}\mu_i = \frac{\mu_i}{\mu_i + \mu_j},$$

and the proof of Claim 18 is complete. $\qquad\square$

By Claim 18, we see that the probabilities that $\mathcal{A}_c$ return arms are with the same form as the MNL model. For a ranking algorithm $\mathcal{A}$, we substitute the input with these $n$ arms and use the procedure $\mathcal{A}_c$ to imitate the comparisons. Whenever the algorithm wants a comparison over $C_i$ and $C_j$, we call procedure $\mathcal{A}_c$ with input $C_i$ and $C_j$. If $\mathcal{A}_c$ returns $C_i$, then we tell $\mathcal{A}$ that $C_i$ wins the comparison, and otherwise, tell $\mathcal{A}_c$ that $C_j$ wins the comparison. Since $\mathcal{A}_c$ returns the arms with probabilities with the same form as the MNL model, $\mathcal{A}$ does not notice any abnormal and work as usual.

For each call of $\mathcal{A}_c$, there is exactly one head generated. Thus, by this modification, $\mathcal{A}$ arranges these $[n]$ coins in the order of ascending head probabilities with confidence $1 - \delta$, and generates $M$ heads in expectation.

This completes the proof of Lemma 6. $\qquad\square$

### B.6   Proof of Proposition 7

**Proposition 7** (Listwise ranking with negligible noises)**.** *If all comparisons are correct w.h.p., to exactly rank $n$ items w.h.p. by using $m$-wise comparisons, $\Theta(n \log_m n)$ comparisons are needed.*

***Proof.*** **Lower Bound.** The proof of the lower bound leverages techniques from information theory. Let $X, Y$ be two discrete random variables (i.e., with at most countably infinite choices of values), and $\Omega_X, \Omega_Y$ be their sample spaces, respectively. We first briefly introduce some terms of information theory. More information about the information theory can be found in standard texts (e.g., [9]).

Define

$$p_x := \mathbb{P}\{X = x\}, \quad p_y := \mathbb{P}\{Y = y\},$$
$$p_{x,y} := \mathbb{P}\{X = x, Y = y\}, \quad p_{x|y} := \mathbb{P}\{X = x \mid Y = y\}.$$

The information entropy of $X$ is defined as

$$H(X) := \sum_{x \in \Omega_X} p_x \log(1/p_x),$$

and the information entropy of $Y$ is defined as

$$H(Y) := \sum_{y \in \Omega_Y} p_y \log(1/p_y).$$

The joint entropy of $X$ and $Y$ is

$$H(X, Y) := \sum_{x \in \Omega_X, y \in \Omega_Y} p_{x,y} \log(1/p_{x,y}).$$

The conditional entropy of $X$ given $Y = y$ is

$$H(X \mid Y = y) := \sum_{x \in \Omega_X} p_{x|y} \log(1/p_{x|y}),$$

and the conditional entropy of $X$ given $Y$ is

$$H(X \mid Y) := \sum_{y \in \Omega_Y} p_y H(X \mid Y = y).$$

The mutual information of $X$ and $Y$ is

$$I(X; Y) := \sum_{x \in \Omega_X, y \in \Omega} p_{x,y} \log \frac{p_{x,y}}{p_x p_y}.$$

Given another discrete random variable $Z$, the conditional mutual information of $X$ and $Y$ given $Z$ is

$$I(X; Y \mid Z) := I(X; Y, Z) - I(X; Z).$$

We further have the following facts [9]

$$H(X) \leq \log |\Omega_X|,$$
$$H(X \mid Y) \leq H(X) \leq H(X, Y),$$
$$H(X, Y) = H(Y) + H(X \mid Y) = H(X) + H(Y \mid X),$$
$$I(X; Y) = H(X) - H(X \mid Y),$$
$$I(X; Y \mid Z) \leq I(X; Y).$$

Also, if $X$ is determined by $Y$, then

$$H(X \mid Y) = 0.$$

With the above introduction of information, we show the following fact that is used in the proof.

**Fact 19** (Fano's Inequality [14])**.** *To recover the value of $X$ from $Y$ with error probability no more than $\delta$, it must hold that*

$$H(X|Y) \leq H(\delta) + \delta \log(|\Omega_X| - 1).$$

The key idea to prove the lower bound is to show that if the expected number of samples conducted is lower than the lower bound, then Fano's Inequality will not be satisfied.

From now on, we assume that all the comparisons are correct and choose $\delta = 1/4$. We reuse some notation and let $X$ be the ranking of the $n$ items. Before any comparison, we have no information about it, and thus, each ranking has the same probability to be the correct one. Since there are $n!$ possible permutations in total, we have that $H(X) = \log(n!) \simeq n \log n$.

Let $\mathcal{A}$ be an algorithm that adaptively selects the sets to compare and determine whether to stop by past comparison outcomes, let $N$ be the number of comparisons conducted till termination (i.e., stopping time). Let $\vec{S} = (S_1, S_2, ..., S_N)$ be the sequence of sets that the algorithm compares. Let $\vec{Y} = (Y_1, Y_2, ..., Y_N)$ be the sequence of comparison outcomes generated by the algorithm. For

any $t$, $S_t$ is of the form $(S_t[1], S_t[2], ..., S_t[m])$, which consists of the items compared in the $t$-th comparison. The value of $Y_t$ is in $\{1, 2, ..., m\}$, where $Y_t = i$ means the winner of the $t$-th comparison is $S_t[i]$. We assume that $\mathcal{A}$ is deterministic, i.e., the value of $S_t$ is determined by $(Y_1, Y_2, ..., Y_{t-1})$ and $(S_1, S_2, ..., S_{t-1})$, and $N$ is determined by $\vec{Y}$ and $\vec{S}$. We have

$$
\begin{aligned}
I(X; \vec{S}|\vec{Y}, N) \leq & I(X; \vec{S} \mid \vec{Y}) \leq H(\vec{S}|\vec{Y}) \\
= & H(S_1 \mid \vec{Y}) + H(S_2 \mid \vec{Y}) + \cdots H(S_N \mid \vec{Y}) \\
\leq & H(S_1) + H(S_2 \mid Y_1,) + \cdots H(S_N \mid Y_1, Y_2, ..., Y_{N-1}) \\
= & 0.
\end{aligned}
\tag{10}
$$

Also, for any $t$-th comparison, there are at most $m$ different choices of values for $Y_t$, and thus, $H(Y_t) \leq \log m$. For any $n \in \mathbb{Z}^+$, when $N = n$, the number of choices of values of $\vec{Y}$ is at most $m^n$, so $H(\vec{Y}|N = n) \leq n \log m$, which implies that

$$
H(\vec{Y}|N) = \sum_{n=1}^{\infty} \mathbb{P}\{N = n\} H(\vec{Y} \mid N = n) \leq \mathbb{E}N \log m.
\tag{11}
$$

Now, we bound $H(N)$ by $\mathbb{E}N$. Define a random variable $R$ such that $R = 0$ if $N < 2\mathbb{E}N$ and $R = k$ if $2^k \mathbb{E}N \leq N < 2^{k+1}\mathbb{E}N$ for any $k \in \mathbb{Z}^+$. By Markov's Inequality, we have that for $k \in \mathbb{Z}^+$,

$$
\mathbb{P}\{R = k\} = \mathbb{P}\{2^k \mathbb{E}N \leq N < 2^{k+1}\mathbb{E}N\} \leq \mathbb{P}\{N \geq 2^k \mathbb{E}N\} \leq 2^{-k},
\tag{12}
$$

Use $p_k$ to denote $\mathbb{P}\{R = k\}$. By analyzing the function $p \log(1/p), p \in [0, 1]$, it holds that

$$
H(R) = p_0 \log(1/p_0) + \sum_{k=1}^{\infty} p_k \log(1/p_k) \leq 2/e + \sum_{k=2}^{\infty} 2^{-k} \log(2^k) \leq 2/e + (3/2)\log 2.
\tag{13}
$$

Noting that $H(N \mid N \in S) \leq \log|S|$ for all sets $S$, we have

$$
\begin{aligned}
H(N) = & H(R) + H(N|R) \\
= & H(R) + \mathbb{P}\{N < 2\mathbb{E}N\} H(N \mid N < 2\mathbb{E}N) \\
& + \sum_{k=1}^{\infty} \mathbb{P}\{2^k \mathbb{E}N \leq N < 2^{k+1}\mathbb{E}N\} H(N \mid 2^k \mathbb{E}N \leq N < 2^{k+1}\mathbb{E}N) \\
\overset{(a)}{\leq} & 2/e + (3/2)\log 2 + \log(2\mathbb{E}N) + \sum_{i=1}^{\infty} 2^{-k} \log(2^k \mathbb{E}N) \\
\leq & 2/e + \log(24\mathbb{E}^2 N),
\end{aligned}
\tag{14}
$$

where (a) is due to (12) and (13).

By (10) (11) (14), we have

$$
\begin{aligned}
H\left(X \mid N, \vec{Y}, \vec{A}\right) = & H(X) - I\left(X; N, \vec{Y}, \vec{A}\right) \\
\geq & H(X) - H\left(N, \vec{Y}, \vec{A}\right) \\
= & H(X) - \left(H(N) + H(\vec{Y} \mid N) + H(\vec{A} \mid N, \vec{Y})\right) \\
\geq & \log(n!) - \left(2/e + \log(24\mathbb{E}^2 N) + \mathbb{E}N \log m + 0\right).
\end{aligned}
\tag{15}
$$

By Fano's Inequality, to recover $X$ with probability at least $1/4$, it must hold that

$$
H(X \mid N, \vec{Y}, \vec{A}) \leq H(1/4) + (1/4)\log(n! - 1),
$$

which, along with (15) and $\log(n!) = \Theta(n \log n)$, implies that

$$
\mathbb{E}N = \Omega(n \log_m n).
$$

---

**Algorithm 6** ListwiseMerge($A_1, A_2, ..., A_m, m$)

---

1: $Ans \leftarrow$ an empty list to store the result;
2: For all $i$ in $[m]$, Let $I_i \leftarrow 1$ be the index of $A_i$;
3: **while** $\exists i \in [m], I_i \leq |A_i|$ **do**
4:      $B \leftarrow \{A_i[I_i] : I_i \leq A_i\}$;
5:      Conduct a listwise comparison over $B$, and let $A_j[I_j]$ be the winner;
6:      Push $A_j[I_j]$ to the end of $Ans$; $I_j \leftarrow I_j + 1$;
7: **end while**
8: **return** $Ans$

---

---

**Algorithm 7** ListwiseMergeSort($S, m$) (LWMS)

---

1: **if** $|S| = 1$ **then**
2:      **return** $S$;    # No need to do anything
3: **end if**
4: Divide $S$ into $m$ sets $A_1, A_2, ..., A_3$ such that $|A_i| \leq \lceil |S|/m| \rceil$ for all $i \in [m]$;
5: **for** $i \in [m]$ **do**
6:      $A_i \leftarrow$ ListwiseMergeSort($A_i, m$)
7: **end for**
8: **return** ListwiseMerge($A_1, A_2, ..., A_m, m$);

---

For randomized algorithms, its sample complexity is no less than that of the fastest deterministic algorithm, and thus, satisfies the same lower bound. This proves the lower bound.

**Upper Bound.** To see the upper bound, consider the following ListwiseMergeSort (LWMS) algorithm, which is presented in Algorithm 7. LWMS is similar to the binary merge-sort. Algorithm 6 ListwiseMerge is the subroutine of LWMS, which merges $m$ sorted lists of items.

**Lemma 20** (Theoretical upper bound of LWMS). *Algorithm LWMS correctly ranks $n$ items with high probability using $O(n \log_m n)$ comparisons.*

*Proof.* We use $T_s(x)$ to denote the number of comparisons needed to rank (sort) $x$ items, and use $T_m(x)$ to denote the number of comparisons needed to merge $m$ sorted lists with $x$ items in total. In the algorithm ListwiseMerge, since after each comparison, a new item is added to the result $Ans$, we have that $T_m(x) \leq x$. Also, we have that $T_s(1) = 0$, and for all $t \geq 1$, $T_s(m^t) = mT_s(m^{t-1}) + T_m(m^t)$. It then follows that $T_s(m^t) \leq tm^t$, which implies $T_s(n) = O(n \log_m n)$. This completes the proof. $\square$

This completes the proof of Proposition 7. $\square$

## B.7   Proof of Theorem 8

**Theorem 8.** *Under the MNL model, given $n$ items with preference scores $\theta_1, \theta_2, ..., \theta_n$ and $\Delta_{i,j} := |\theta_i/(\theta_i + \theta_j) - 1/2|$, $\tilde{\Delta}_i = \Delta_i := \min_{j \neq i} \Delta_{i,j}$, to correctly rank these $n$ items with probability $1 - \delta$, even with $m$-wise comparisons for all $m \in \{2, 3, ..., n\}$, the lower bound is the same as the pairwise ranking (i.e., Theorem 3) with (possibly) different hidden constant factors.*

*Proof.* Let $n$ coins $C_1, C_2, ..., C_n$ with unknown head probabilities $\mu_1, \mu_2, ..., \mu_n$ be given, where $\mu_i/\theta_i$ is a fixed constant for all $i \in [n]$. We only need to show that the reduction from PEMAB problems to exact ranking stated in Lemma 6 still holds for listwise comparisons under the MNL model. $\square$

Consider the the following procedure:

**Claim 21.** *Procedure $\mathcal{A}'_c$ returns a coin $C_{r_i}$ with probability $\mu_{r_i}/\sum_{j=1}^{m} \mu_{r_j}$.*

---
**Algorithm** Procedure $\mathcal{A}'_c$
___
**Input:** Totally $m$ coins $C_{r_1}, C_{r_2}, ..., C_{r_m}$ with unknown head probabilities $\mu_{r_1}, \mu_{r_2}, ..., \mu_{r_m}$;

1: **repeat**
2:     Randomly choose a coin $C_w$ and toss it;
3:     Let $s \leftarrow 1$ if the toss gives a head, and let $s \leftarrow 0$ otherwise;
4: **until** $s = 1$
5: **return** $C_w$;
___

***Proof of Claim 21.*** Let $T$ be the number of tosses conducted before $\mathcal{A}'_c$ returns, and $X$ be the coin $\mathcal{A}'_c$ returns. By using conditional probability, we have that for all $t \geq 1$ and $i$ in $[m]$,

$$\mathbb{P}\{T = t, X = C_{r_i}\} = \prod_{\tau=1}^{t-1} \mathbb{P}\{T > \tau \mid T > \tau - 1\} \cdot \mathbb{P}\{T = t, X = C_{r_i} \mid T > t - 1\}$$

$$= (\mathbb{P}\{T > 1\})^{t-1} \mathbb{P}\{T = 1, X = C_{r_i}\}$$

$$= \left(1 - \frac{1}{m}\sum_{j=1}^m \mu_{r_j}\right)^{t-1} \cdot \frac{1}{m}\mu_{r_i},$$

$$\mathbb{P}\{X = C_{r_i}\} = \sum_{t=1}^{\infty} \mathbb{P}\{T = t, X = C_{r_i}\}$$

$$= \sum_{t=1}^{\infty} \left(1 - \frac{1}{m}\cdot\sum_{j=1}^m \mu_{r_j}\right)^{t-1} \cdot \frac{1}{m}\mu_{r_i} = \frac{\mu_{r_i}}{\sum_{j=1}^m \mu_{r_j}},$$

and the proof of the claim is complete. $\square$

The proof of Theorem 8 is complete by Lemma 5 and the same steps as in the proof of Theorem 3, the pairwise lower bound for the MNL model.

### B.8    Proof of Lemma 9

**Lemma 9** (Theoretical Performance of ATC). *ATC terminates after at most $b^{max} = O(\epsilon^{-2}\log(1/\delta))$ comparisons and returns the more preferred item with probability at least $1/2$. Further, if $\epsilon \leq \Delta_{i,j}$, then ATC returns the more preferred item with probability at least $1 - \delta$.*

***Proof.*** Without loss of generality, we assume $i \succ j$. Since the for loop runs at most $b^{max} = \lceil \frac{1}{2}\epsilon^{-2}\log(2\delta^{-1}) \rceil$ iterations and each iteration performs one comparison, the subroutine returns after at most $O(\epsilon^{-2}\log\delta^{-1})$ comparisons. Since the return condition of items $i$ and $j$ are symmetric and $i \succ j$, by this symmetry, ATC returns $j$ with probability no more than $1/2$.

Now we consider the case where $p_{i,j} \geq 1/2 + \epsilon$, and it remains to prove that ATC returns $i$ with probability at least $1 - \delta$. Define $b^t := \sqrt{\frac{1}{2t}\log\frac{\pi^2 t^2}{3\delta}}$. Let $\mathcal{E}_t^{out}$ be the event that $\hat{p}_i^t \leq p_{i,j} - b^t$, and define $\mathcal{E}^{out} := \bigcup_{t=1}^{\infty} \mathcal{E}_t^{out}$. We have

$$\mathbb{P}\{\mathcal{E}^{out}\} \overset{(a)}{\leq} \sum_{t=1}^{\infty} \mathbb{P}\{\mathcal{E}_t^{out}\} \overset{(b)}{\leq} \sum_{t=1}^{\infty}\left[\exp\left(-2t\left(b^t\right)^2\right)\right] \leq \sum_{t=1}^{\infty}\frac{3\delta}{\pi^2 t^2} \leq \frac{\delta}{2}, \qquad (16)$$

where (a) is due to the union bound and (b) is due to the Chernoff-Hoeffding Inequality [19].

Assume that $\mathcal{E}^{out}$ does not happen, and we have that for all $t$, $\hat{p}_i^t > 1/2 + \epsilon - b^t \geq 1/2 - b^t$. Thus, ATC does not return $j$ during the for loop with probability at least $1 - \delta/2$.

After the for loop, by Chernoff-Hoeffding Inequality and $b^{max} = \lceil\frac{1}{2\epsilon^2}\log\frac{2}{\delta}\rceil$, we have

$$\mathbb{P}\left\{\hat{p}_i^{b^{max}} \leq 1/2\right\} \leq \exp\left\{-2b^{max}(p_{i,j} - 1/2)^2\right\} \leq \exp\left\{-2b^{max}\epsilon^2\right\} \leq \delta/2, \qquad (17)$$

which implies that the last line of ATC returns $i$ with probability at least $1 - \delta/2$. This completes the proof of Lemma 9. $\square$

## B.9 Proof of Lemma 10

**Lemma 10** (Theoretical performance of ATI). *Let $\delta \in (0,1)$. ATI returns after $O(\epsilon^{-2}\log(|S|/\delta))$ comparisons and, with probability at least $1-\delta$, correctly inserts $i$ or returns unsure. Further, if $\epsilon \le \Delta_i$, it correctly inserts $i$ with probability at least $1-\delta$.*

*Proof.* **(I)** We first prove the sample complexity. We observe that for a constant $\delta_0 \in (0,1/2)$, a call of ATC$(i,j,\epsilon,\delta_0)$ returns after at most $O(\epsilon^{-2})$ comparisons by Lemma 9. In ATI, for each iteration, there are at most three calls of ATC and all the calls are with constant confidence. Also, ATI returns after at most $t^{max} = O(h + \log \delta^{-1})$ iterations, where $h = 1 + \lceil \log_2(1 + |S|) \rceil = O(\log |S|)$. Thus, the number of comparisons is at most $3t^{max} \cdot O(\epsilon^{-2}) = O(\epsilon^{-2}\log(|S|/\delta))$. This completes the proof sample complexity.

**(II)** We prove that ATI does not insert $i$ into a wrong place with probability at least $1/2$. A round (or iteration) is said to be *correct* if during this round, all calls of ATC return the more preferred item, and is said to be *incorrect* otherwise. A leaf node $u$ is said to be *correct* if $i \in (u.\text{left}, u.\text{right})$, i.e., $i$ belongs to the corresponding interval of $u$. A leaf node $u$ is said to be *incorrect* if it is not correct.

For any round $t$, we define an event $\mathcal{E}_{il}^t$ such that

$$\mathcal{E}_{il}^t := \{X = \text{some incorrect leaf node at the beginging of round } t \text{ and } c_X \ge 1\}. \qquad (18)$$

We assume that for some round $t$, $\mathcal{E}_{il}^t$ happens, which implies that $i \succ u.\text{right}$ or $u.\text{left} \succ i$, i.e., $i$ does not belong to the interval of $u$. By Lemma 9 the property of ATC, it holds that

$$\mathbb{P}\{\text{ATC}(i, u.\text{right}, \epsilon, \delta) = i \mid i \succ u.\text{right}\} \ge 1/2,$$
$$\mathbb{P}\{\text{ATC}(i, u.\text{left}, \epsilon, \delta) = u.\text{left} \mid u.\text{left} \succ i\} \ge 1/2.$$

which implies that for any round $t$,

$$\mathbb{P}\{\text{round } t \text{ is correct} \mid \mathcal{E}_{il}^t\} \ge 1/2. \qquad (19)$$

For any $t$, define

$$R_1^t := |\{\tau \le t : \text{round } \tau \text{ is correct, and } \mathcal{E}_{il}^\tau \text{ happens}\}|,$$
$$W_1^t := |\{\tau \le t : \text{round } \tau \text{ is incorrect, and } \mathcal{E}_{il}^\tau \text{ happens}\}|.$$

For any incorrect leaf node $u$ and any round $t$, the counter $c_u$ is increased by one during this round if and only if $\mathcal{E}_{il}^\tau$ happens and this round is incorrect. Also, for any round $t$, given $\mathcal{E}_{il}^\tau$, if this round is correct, then the counter $c_u$ is decreased by one. Thus, for any incorrect leaf node $u$, at the end of any round $t$, the value of $c_u$ is at most

$$c_u(t) \le 1 + W_1^t - R_1^t.$$

After the for loop, ATI incorrectly inserts $i$ if and only if some incorrect leaf node $u$ is counted for $\frac{5}{16}t^{max} + 1$ times, i.e., $c_u \ge \frac{5}{16}t^{max} + 1$, which implies $W_1^{t^{max}} - R_1^{t^{max}} \ge \frac{5}{16}t^{max}$. Thus, by the fact that $W_1^{t^{max}} + R_1^{t^{max}} \le t^{max}$, and Eq. 19, we obtain

$$\mathbb{P}\left\{W_1^{t^{max}} - R_1^{t^{max}} \ge \frac{5}{16}t^{max}\right\}$$

$$= \mathbb{P}\left\{W_1^{t^{max}} \ge \frac{1}{2}\left(R_1^{t^{max}} + W_1^{t^{max}} + \frac{5}{16}t^{max}\right)\right\}$$

$$\overset{(a)}{\le} \sup_{K \le t^{max}} \mathbb{P}\left\{\frac{W_1^{t^{max}}}{W_1^{t^{max}} + R_1^{t^{max}}} \ge \frac{1}{2} + \frac{5}{32} \cdot \frac{t^{max}}{W_1^{t^{max}} + R_1^{t^{max}}} \,\Big|\, W_1^{t^{max}} + R_1^{t^{max}} = K\right\}$$

$$\overset{(b)}{\le} \sup_{K \le t^{max}} \exp\left\{-2K\left(\frac{5t^{max}}{32K}\right)^2\right\}$$

$$= \exp\left\{-2t^{max}\left(\frac{5t^{max}}{32t^{max}}\right)^2\right\} \le \delta/2, \qquad (20)$$

where (a) is due to $R_1^{t^{max}} + W_1^{t^{max}} \leq t^{max}$, and (b) follows from Chernoff-Hoeffding Inequality. This proves that with probability at least $1 - \delta/2$, $i$ is not inserted into a wrong place by the second last line.

Then, during the for loop, for any $t \leq t^{max}$, by (19) and Chernoff-Hoeffding Inequality, we have that at the end of the $t$-th round, the probability that $X$ equals to an incorrect leaf node and $c_X > \frac{1}{2}t + \sqrt{\frac{t}{2} \log \frac{\pi^2 t^2}{3\delta}} + 1$ is at most

$$
\mathbb{P}\left\{ W_1^t - R_1^t \geq \frac{1}{2}t + \sqrt{\frac{t}{2} \log \frac{\pi^2 t^2}{3\delta}} \right\} \leq \mathbb{P}\left\{ W_1^t \geq \frac{1}{2}t + \sqrt{\frac{t}{2} \log \frac{\pi^2 t^2}{3\delta}} \right\}
$$

$$
\leq \exp\left\{ -\frac{2}{t}\left( \frac{1}{2}t + \sqrt{\frac{t}{2} \log \frac{\pi^2 t^2}{3\delta}} - \frac{t}{2} \right)^2 \right\} \leq \frac{3\delta}{\pi^2 t^2}.
$$

Since

$$
\sum_{t=1}^{\infty} \frac{3\delta}{\pi^2 t^2} \leq \delta/2,
$$

during the for loop, with probability at least $1 - \delta/2$, ATI does not insert $i$ into a wrong place. This, along with Eq. (20), proves that with probability at least $1 - \delta$, ATI does not insert $i$ into a wrong place. This completes the proof of the first part of Lemma 10.

**(III)** In this part, we assume $\epsilon \leq \Delta_i$ and we prove the second part of Lemma 10. For any round $t$, by Lemma 9 and the choice of input parameters of the calls of ATC, this round is correct with probability at least $q$. Here, we define $R$ as the number of correct rounds before termination, and let $W$ be the number of incorrect rounds before termination.

Let $u_0$ be the correct node. Define the distance between two nodes $u$ and $v$ as $d(u, v) :=$ the length of the shortest path from $u$ to the $v$, i.e., the number of edges between $u$ and $v$. During each correct round, either $d(X, u_0)$ is decreased by one or the value of $c_{u_0}$ is increased by one, i.e., $c_{u_0} - d(X, u_0)$ is increased by one. During each incorrect round, either $d(X, u_0)$ is increased by one or the value of $c_{u_0}$ is decreased by one, i.e., $c_{u_0} - d(X, u_0)$ is decreased by one. Since the distance between the start node (i.e., the root node) and $u_0$ is at most $h - 1$, we always have

$$
R - W \leq h - 1 + (c_{u_0} - d(X, u_0)).
$$

After the for loop, if $c_{u_0} \geq \frac{5}{16}t^{max} + 1$, then ATI correctly inserts $i$. Thus, if $R - W \geq h + \frac{5}{16}t^{max}$, then ATI correctly inserts $i$.

Assume that ATI does not return during the for loop, and then, we have $R + W = t^{max}$. For all $t$, round $t$ is correct with probability at least $q$ by Lemma 9 and the choices of input parameters of the calls of ATC, hence, by $t^{max} \geq \max\{4h, \frac{512}{25} \log \frac{2}{\delta}\}$ and $q = 15/16$, we have

$$
\mathbb{P}\left\{ R - W < h + \frac{5}{16}t^{max} \right\} \overset{(a)}{\leq} \mathbb{P}\left\{ R - W < \left( \frac{1}{4} + \frac{5}{16} \right) t^{max} \right\}
$$

$$
= \mathbb{P}\left\{ R - (t^{max} - R) < \left( \frac{1}{4} + \frac{5}{16} \right) t^{max} \right\}
$$

$$
= \mathbb{P}\left\{ R < \frac{25}{32}t^{max} \right\}
$$

$$
\overset{(b)}{\leq} \exp\left\{ -2t^{max}\left( q - \frac{25}{32} \right)^2 \right\} \leq \frac{\delta}{2},
$$

where (a) is due to $t^{max} \geq 4h$ and (b) follows from Chernoff-Hoeffding Inequality.

In conclusion, when $\epsilon \leq \Delta_i$, if ATI does not return during the for loop, then it will, with probability at least $1 - \delta/2$, insert $i$ into a correct position by the second last line (after the for loop). Also, by part (II), with probability at least $1 - \delta/2$, ATI does not insert $i$ into a wrong position during the for loop. Thus, when $\epsilon \leq \Delta_i$, ATI correctly inserts the input item $i$ with probability at least $1 - \delta$. This proves the second part of Lemma 10, and along with parts (I) and (II), completes the proof. $\qquad\square$

## B.10 Proof of Lemma 11

**Lemma 11** (Theoretical Performance of IAI). *With probability at least $1 - \delta$, IAI correctly inserts $i$ into $S$, and conducts at most $O(\Delta_i^{-2}(\log\log\Delta_i^{-1} + \log(|S|/\delta)))$ comparisons.*

*Proof.* Define events

$$
\mathcal{E}_1^t :=\{\epsilon_t > \Delta_i \text{ and IAI does not insert } i \text{ into a wrong position}\},
$$
$$
\mathcal{E}_2^t :=\{\epsilon_t \leq \Delta_i \text{ and IAI correctly inserts } i\},
$$

and the bad event

$$
\mathcal{E}^{bad} := \bigcup_{t=1}^{\infty}(\mathcal{E}_1^t \cup \mathcal{E}_2^t)^{\complement}.
$$

By the union bound and Lemma 10, we have

$$
\mathbb{P}\{\mathcal{E}^{bad}\} \leq \sum_{t=1}^{\infty}\mathbb{P}\left\{\left(\mathcal{E}_1^t \cup \mathcal{E}_2^t\right)^{\complement}\right\} \leq \sum_{t=1}^{\infty}\delta_t = \sum_{t=1}^{\infty}\frac{6\delta}{\pi^2 t^2} = \delta.
$$

In this proof, we assume that $\mathcal{E}^{bad}$ does not happen.

**Correctness.** We first prove the correctness. By the definition of $\mathcal{E}^{bad}$, for all $t$ such that $\epsilon_t > \Delta_i$, IAI does not insert $i$ into a wrong position, and when $\epsilon_t \leq \Delta_i$, IAI correctly inserts $i$. Since $\lim_{t\to\infty}\epsilon_t = 0$, there is a $t^*$ such that $\epsilon_{t^*} \leq \Delta_i$. Thus, when $\mathcal{E}^{bad}$ does not happen, IAI correctly inserts $i$. Since $\mathcal{E}^{bad}$ happens with probability at most $\delta$, the correctness follows.

**Sample complexity**. Second, we prove the sample complexity. Let $\tau$ be the integer such that $\epsilon_\tau \leq \Delta_i < \epsilon_{\tau-1}$. By the definition of $\mathcal{E}^{bad}$, when $\mathcal{E}^{bad}$ does not happen, IAI correctly inserts $i$ and returns before the end of the $\tau$-th round.

By $\epsilon_{\tau-1} = 2^{-\tau}$ and $\epsilon_{\tau-1} > \Delta_i$, we have $\tau < \log_2\Delta_i^{-1}$. For $1 \leq t \leq \tau$, by Lemma 10, the $t$-th round of IAI conducts at most $O(\epsilon_t^{-2}\log(|S| \cdot \delta_t^{-1}))$ comparisons. Thus, given $\mathcal{E}^{bad}$ does not happen, the number of comparisons conducted by IAI is at most

$$
\begin{aligned}
O\left(\sum_{t=1}^{\tau}\epsilon_t^{-2}\log\left(|S|/\delta_t\right)\right) &\overset{(a)}{=} O\left(\sum_{t=1}^{\tau}\left(2^{t+1}\right)^2\log\left(\pi^2 t^2 |S|/(6\delta)\right)\right) \\
&= O\left(\sum_{t=1}^{\tau}4^t \cdot \log\left(|S|\tau/\delta\right)\right) \\
&= O(4^\tau \cdot \log\left(|S|\tau/\delta\right)) \\
&\overset{(b)}{=} O\left(4^{\log_2(1/\Delta_i)} \cdot \log(|S| \cdot \log(1/\Delta_i)/\delta)\right) \\
&= O\left(\Delta_i^{-2}\left(\log\log\Delta_i^{-1} + \log\left(|S|/\delta\right)\right)\right),
\end{aligned}
$$

where (a) follows from $\epsilon_t = 2^{t+1}$ and $\delta_t = \frac{6\delta}{\pi^2 t^2}$, and (b) is due to $\tau < \log_2(1/\Delta_i)$. This proves the sample complexity.

The proof of Lemma 11 is complete. $\square$

## B.11 Proof of Theorem 12

**Theorem 12** (Theoretical Performance of IIR). *With probability at least $1 - \delta$, IIR returns the exact ranking of $[n]$ and conducts at most $O(\sum_{i\in[n]}\Delta_i^{-2}(\log\log\Delta_i^{-1} + \log(n/\delta)))$ comparisons.*

*Proof.* At iteration $t$ for each $t \in \{2, 3, ..., n\}$, by Lemma 11, with probability at least $1 - \delta/(n-1)$, the call of IAI correctly inserts $S[t]$ into $Ans$, and uses at most $O(\Delta_{S[t]}^{-2}(\log\log\Delta_{S[t]}^{-1} + \log(n/\delta)))$ comparisons. The desired sample complexity follows by summing up the upper bounds for $t \in \{2, 3, ..., n\}$. For correctness, if all calls of IAI are correct (which happens with probability at least $1 - \delta$ by the union bound), then IIR correctly returns the true ranking. This completes the proof. $\square$

### B.12 Proof of Lemma 15

**Lemma 15** (Lower bound for ranking two items). *Let $\delta \in (0, 1/4)$ and $\delta$-correct algorithm $\mathcal{A}_2$ be given. Let $T_{\mathcal{A}_2}(\Delta_{i,j})$ be the number of comparisons conducted by $\mathcal{A}_2$ under the $\Delta_{i,j}$-values. To rank $i$ and $j$ with error probability no more than $\delta$, there is a universal constant $c_{lb2} > 0$ such that*

$$\limsup_{\Delta_{i,j} \to 0} \frac{\mathbb{E}[T_{\mathcal{A}_2}(\Delta_{i,j})]}{\Delta_{i,j}^{-2}(\log \log \Delta_{i,j}^{-2} + \log \delta^{-1})} \geq c_{lb2}. \tag{3}$$

*Proof.* We will invoke the results for pure exploration multi-armed bandit (PEMAB) problems, and we refer to [27] as a reference for details about PEMAB. Assume that there is an arm $a$, and whenever it is pulled for the $t$-th time, it gives an i.i.d. reward $Y^t$. Further assume that for $t \in \mathbb{Z}^+$, $Y^t$ is a Gaussian random variable with mean $\eta$ and variance 1. We assume that $|\eta| \leq 1/2$ and $\eta \neq 0$. Let $\mathcal{B}$ be a $\delta$-correct algorithm that has no knowledge of $\eta$ and is able to tell whether $\eta > 0$ with probability $1 - \delta$ for any non-zero $\eta$-value. Let $T_{\mathcal{B}}(\eta)$ be the number of pulls $\mathcal{B}$ uses before termination under the given $\eta$-value. The authors of [20, 15] have shown that

$$\limsup_{|\eta| \to 0} \frac{\mathbb{E}[T_{\mathcal{B}}(\eta)]}{\eta^{-2} \log \log \eta^{-2}} \geq 2 - 4\delta. \tag{21}$$

In this proof, we reduce the problem of distinguishing whether $\eta > 0$ to the problem of ranking two items. For any $t \in \mathbb{Z}^+$, if $0 < \eta < 1/2$, we have

$$\mathbb{P}\{Y^t \geq 0\} = \int_{-\eta}^{\infty} e^{-\frac{x^2}{2}} \mathrm{d}x \geq \frac{1}{2} + \eta \cdot e^{-\frac{\eta^2}{2}} \geq \frac{1}{2} + \eta \cdot e^{-1/8},$$

and if $-1/2 < \eta < 0$, we have

$$\mathbb{P}\{Y^t < 0\} = \int_{-\infty}^{-\eta} e^{-\frac{x^2}{2}} \mathrm{d}x \geq \frac{1}{2} + |\eta| \cdot e^{-\frac{\eta^2}{2}} \geq \frac{1}{2} + |\eta| \cdot e^{-1/8}.$$

For each $t$, we let $Z^t = 2 \cdot \mathbb{1}\{Y^t \geq 0\} - 1$. When $\eta > 0$, $Z^t$ is with probability at least $1/2 + \eta \cdot e^{-1/8}$ to be 1, and when $\eta < 0$, it is with probability at least $1/2 + |\eta| \cdot e^{-1/8}$ to be $-1$. Thus, we can view that $(Z^t, t \in \mathbb{Z}^+)$ are generated by tossing a coin with $\mathbb{P}\{Y^t \geq 0\}$ head probability, and we have $|\mathbb{P}\{Y^t \geq 0\} - 1/2| \geq e^{1/8}|\eta|$. Assume $\mathcal{A}_2$ can ranking two items $i$ and $j$ with probability $1 - \delta$ by $T_{\mathcal{A}_2}(\Delta_{i,j})$ expected number of comparisons, then it can find whether $\eta > 0$ by at most $T_{\mathcal{A}_2}(\eta \cdot e^{-1/8})$ expected number of pulls of the arm $a$. Thus, by (21), we have

$$\limsup_{\Delta_{i,j} \to 0} \frac{\mathbb{E}[T_{\mathcal{A}_2}(\Delta_{i,j})]}{\Delta_{i,j}^{-2} \log \log \Delta_{i,j}^{-2}} \geq e^{-1/4}(2 - 4\delta). \tag{22}$$

Then, by the previous work [27], we obtain another lower bound on ranking two items, i.e., $\Omega(\Delta_{i,j}^{-2} \log \delta^{-1})$. Summing up this lower bound and (22), we obtain the desired lower bound. This completes the proof. $\qquad \square$

### B.13 Proof of Lemma 16

**Lemma 16** (Reductions). *With the above definitions, (i) if the true ranking of $[n]$ is found, with no more comparisons, one can get the solution of $\mathcal{P}_1$, and (ii) if an algorithm solves $\mathcal{P}_1$ with $N$ expected number of comparison, there is another algorithm that solves $\mathcal{P}_2$ with $N$ expected number of tosses.*

*Proof.* We first prove the reduction from $\mathcal{P}_1$ to exact ranking. Given an instance of $\mathcal{P}_1$, we simply use an exact ranking algorithm to find its true ranking. By the assumptions made in the construction of $\mathcal{P}_1$, the comparison probabilities under the correct hypothesis $\mathcal{H}_{\vec{\pi}^0}$ is exactly the same as the corresponding ranking instance. Thus, by the found true ranking, we can find the true hypothesis with no more comparisons. This completes the first part of Lemma 16.

Secondly, we prove the reduction from $\mathcal{P}_2$ to $\mathcal{P}_1$. Assume that $n$ is odd and $n = 2m + 1$, and when $n$ is even, we can prove the same results by similar steps. Let $\mathcal{B}$ be an arbitrary $\delta$-correct algorithm for

$\mathcal{P}_1$. Let the $\binom{n}{2}$ coins satisfying the restrictions of $\mathcal{P}_2$ be given. We construct $n$ virtual items indexed by $r_1, r_2, ..., r_n$, where $(r_1, r_2, ..., r_n)$ is a permutation of $[n]$. With these $n$ items, we construct $2^m$ hypotheses as defined in the construction of Problem $\mathcal{P}_1$ (i.e., $\mathcal{H}_{\vec{\pi}}, \vec{\pi} \in \{0,1\}^m$). Then, we send these $n$ items and the hypotheses as the input to algorithm $\mathcal{B}$. Whenever $\mathcal{B}$ wants a comparison over the pair $(r_i, r_j)$, we toss the coin $C_{i,j}$. If the toss gives a head, we tell $\mathcal{B}$ that the winner of the comparison is $r_i$, and if the toss gives a tail, we tell $\mathcal{B}$ that the winner is $r_j$. Since the values of the head probabilities $\mu_{i,j}$ are lawful for the comparison probabilities of Problem $\mathcal{P}_1$, $\mathcal{B}$ does not notice any abnormal and works as usual. Finally, $\mathcal{B}$ terminates and returns a $\vec{\pi} \in \Pi$.

For any $k \in [m]$, if $\pi(k) = 1$, then we return $\mu_{2k-1,2k} > 1/2$, and otherwise, we return $\mu_{2k-1,2k} < 1/2$. If $\mathcal{B}$ returns a correct hypothesis for these $n$ virtual items, one can determine whether $\mu_{2k-1,2k} > 1/2$ for any $k \in [n]$ by no more tosses of coins. Moreover, for any $(i, j)$, the head probability $\mu_{i,j}$ of problem $\mathcal{P}_1$ equals to $p_{r_i, r_j}$, the comparison probability of problem $\mathcal{P}_2$. This completes the second part of Lemma 16. The proof is complete. $\qquad\square$

### B.14 Proof of Lemma 17

**Lemma 17.** *For $\delta \in (0, 1/12)$, the expected number of tosses needed for solving $\mathcal{P}_2$ is at least*

$$\Omega\Big( \sum_{k\in[m]} \Delta_{q_{2k-1},q_{2k}}^{-2} \cdot \log\log \Delta_{q_{2k-1},q_{2k}}^{-1} + \min\{ \sum_{k\in[m]} \Delta_{q_{2k-1},q_{2k}}^{-2} \cdot \log(\delta_k^{-1}) : \sum_{k\in[m]} \delta_k \leq 2\delta\}\Big). \quad (4)$$

*Proof.* In $\mathcal{P}_2$, the tosses of the coins are independent across time and coins. Also, whether one coin has head probability larger than $1/2$ is independent of other coins. Thus, $\mathcal{P}_2$ is simply a problem such that given $m$ coins with head probability not equal to $1/2$, to identify all the coins with head probabilities larger than $1/2$, and the total error probability is no more than $\delta$.

Given a coin with non-$1/2$ head probability, deciding whether the head probability is larger than $1/2$ is equivalent to the problem of ranking two items, as a toss of a coin with head probability $\eta$ can be viewed as a comparison of items $i$ and $j$ with $p_{i,j} = \eta$. Thus, for coin $C_{2k-1,2k}$, to find whether $\mu_{2k-1.2k} > 1/2$ with at most $\delta_k$ error probability, the expected number of tosses is at least

$$c\Delta_{q_{2k-1},q_{2k}}^{-2} (\log\log \Delta_{q_{2k-1},q_{2k}}^{-1} + \log(1/\delta_k)),$$

where $c > 0$ is a universal constant, and here, we note that $|\mu_{2k-1,2k} - 1/2| = \Delta_{q_{2k-1},q_{2k}}$ for any $k \in [m]$ due to the constructions of $\mathcal{P}_1$ and $\mathcal{P}_2$.

Let $\delta_k$ be the error probability incurred by determining whether $\mu_{2k-1,2k} > 1/2$. To solve $\mathcal{P}_2$ with confidence $1 - \delta$, it is necessary that

$$\prod_{k\in[m]} (1 - \delta_k) \geq 1 - \delta.$$

We also have that for $\delta \in (0, 1/2)$,

$$\sum_{k\in[m]} \delta_k \leq - \sum_{k\in[m]} \log(1 - \delta_k) = -\log \prod_{k\in[m]} (1 - \delta_k)$$
$$\leq -\log(1-\delta) = \log(1 + \delta/(1-\delta))$$
$$\leq \delta/(1-\delta) \leq 2\delta.$$

Thus, the lower bound of $\mathcal{P}_2$ is at least

$$\Omega\Big( \sum_{k\in[m]} \Delta_{r_{2k-1},r_{2k}}^{-2} \log\log \Delta_{r_{2k-1},r_{2k}}^{-1} + \min\{ \sum_{k\in[m]} \Delta_{r_{2k-1},r_{2k}}^{-2} \log \delta_k^{-1} : \sum_{k\in[m]} \delta_k \leq 2\delta\}\Big).$$

This completes the proof of Lemma 17. $\qquad\square$