[Reviews · NeurIPS 2019]

Reviewer 1



The paper is not well written. The ideas are not clearly presented. The paper adapts the previous algorithm to extend it from PAC to instance optimal guarantees which has been previously studied for maximum but not ranking. The paper proved the results theoretically and provided experiments evaluating their algorithm with previous works. The paper also doesn't mention the precise applications where this method will be more useful compared to that of PAC one. Post Rebuttal : I am happy with Author's feedback

Reviewer 2



The paper is of high quality and is well written. It is fairly dense in the number of results and contains strong empirical results to support their theory which makes a good paper for NeurIPS in my humble opinion. I did not fully check the proofs but they seem sound. I am also not very familiar with the ranking literature so my next questions are mostly to help me better understand the results and will guide me to adjust my score. * How realistic is to assume that comparisons are independent across items, sets and time? Is this a widely used assumption in the ranking community? From a layman point of view, this may seem a strong assumption as it is not the same as assuming the typical iid sampling, no? If the assumption, as stated, is required to have a well defined problem, then the question translates to whether or not aiming for exact ranking in general is realistic. * I am not fully clear about the notation in line 49. Why is p_{i, j} disregarding "j" and defined as p_{i, S}? This was a bit confusing to me as later we see \Delta_{i, j} but it is not clear to me the role of j in this case. * It seems that the focus is in having a small number of comparisons. However, what is the computational complexity for m-wise comparison in contrast to pairwise? And, how realistic is the worst case scenario? Where it is claimed that the number of m-wise comparisons is of the same order as number of pairwise comparisons. On a minor note: the shorthand MNL in line 76 is not introduced properly.

Reviewer 3



Originality: The paper solves a previously unsolved problem and the methods seem novel. Quality: The paper is technically sound. I still have some questions: 1. What is the purpose of Section 3.1? Since section 3.2 provides a lower bound for MNL model and MNL is a special case of SST, isn't eqn (2) immediately follows from Theorem 3? Theorem 2 is better in eqn(1), but in what case can it be larger than (2)? 2. Why do the 3 plots in Figure 2 have different number of baselines? Significance: Active ranking under few assumptions is an important question, and previous works only solves this under the PAC setting. I think the results are significant. Clarity: Some small typos: 1. It would be good to clarify the implications of the assumptions on line 50-58: for pairwise comparisons, this just means p_{ij}>1/2 for i>j. 2. Line 87: the square is probably inside the parenthesis?

[Author Response · NeurIPS 2019]

**Reviewer #1. 1) Comparisons** with "From PAC to Instance-Optimal Sample Complexity in the Plackett-Luce Model" by Saha and Gopalan (denoted as [SG]). We were not aware of this paper at the time of submission. Thanks for bringing it to our attention. After reading this paper, we found that our work is different from [SG] in the following aspects. (i) Our paper studies full ranking while [SG] focuses on maxing, which is a different problem. (ii) The algorithms in [SG] only work for the PL model, while our algorithm works for a larger class that strictly contains the pairwise PL model as a special case. The algorithms in [SG] depend on the special properties of the PL model, while ours does not. (iii) In [SG], the algorithm checks whether an item is (PAC-)less preferred than another one by comparing the empirical $p_{i,j}$-value and $(1/2 - \epsilon_s)$. In contrast, our algorithm needs to check whether an item can be inserted to a sorted list with enough confidence, which can hardly be done by simply comparing empirical means. For this purpose, we constructed a subroutine ATC to replace the comparisons in Binary Search in [14] and redesigned the random walk procedure to get an attempting version of Binary Search. (iv) We also proved several lower bounds that are significant for understanding the full ranking problem. Further, to the best of our knowledge, we are the first to introduce the $\Omega(\Delta_i^{-2} \log \log \Delta_i^{-1})$ lower bound to the field of active ranking. **2) PAC v.s. exact ranking.** (i) In some applications, we may want to find the exact order, especially in "winner-takes-all" situations. For example, when predicting the winner of an election, we prefer to get the exact result but not the PAC one, as only a few votes can completely change the result. (ii) Our algorithm IIR can utilize unequal noise levels, while previous PAC algorithms may not. For example, Refs [10,11,12] only depend on $n\epsilon^{-2}$. (iii) Studying exact ranking is theoretical significant and helps obtain insights on instance-wise upper and lower bounds. Specifically, the PAC bounds that depend on $\epsilon^{-1}$ will become vacuous as $\epsilon$ shrinks to zero, i.e., the PAC bounds do not reduce to meaningful instance-wise bounds when one pushes $\epsilon$ to 0.

**Reviewer #2. 1) Independence.** (i) We believe assuming independence is realistic. For example, for ranking movies, it is reasonable to define $p_{i,j}$ as the fraction of users that prefer movie $i$ to $j$, and a comparison between $i$ and $j$ refers to sampling a user's preference. The comparisons can be assumed to be independent if the users are randomly sampled. (ii) Independence across time, items, and sets is a common assumption in this field. (iii) If the comparisons were not independent, we might not be able to recover the true preference of the users. For example, if all annotators compare the items based on the comparison results of some user, then the ranking got from these comparisons may not correctly represent the preference of all users. **2) Notation of** $p_{i,j}$. For a comparison over set $S$, we use $p_{i,S}$ to denote the probability that $i$ wins this comparison. When $|S| = 2$, we write $p_{i,j} = p_{i,\{i,j\}}$ to simplify notation, and $\Delta_{i,j}$ is defined as $|p_{i,j} - 1/2|$. **3) Computational complexity.** (i) It is hard to quantify the actual computational complexity of a comparison as it can vary with applications. For example, a comparison may refer to asking an annotator to compare some items, which can take minutes or longer. Also, a comparison may refer to comparing random numerical values in computer simulations, which may only take milliseconds. Further, due to similar reasons, the difference between the computational complexities of pairwise comparisons and listwise comparisons is also hard to quantify. (ii) It is reasonable to assume that an $m$-wise comparison takes $\Omega(m)$ time to pass the items to compare. Thus, by Proposition 7, we get that ranking $n$ items with $m$-wise comparisons takes $\Omega(mn \log_m n)$ time. From this perspective, pairwise ranking uses less time. **4) Worst case scenario.** (i) The MNL model is a widely used model. Thus, the lower bound for this model is significant in its own right. (ii) We analyzed the listwise ranking bounds for several cases not presented in this paper, and found that whether listwise ranking uses less samples depends on how we define the gaps of listwise comparisons. We believe the general listwise ranking problem deserves an independent paper. **5) Comparisons.** Comparisons may refer to querying users about their preferences, asking annotators to compare items, or retrieving data from a data center. We view comparisons as black-box procedures in this paper. **6) Assumptions.** For A1, please refer to "1) Independence" above. If A2 is not true, then the unique true ranking does not exist, rendering the problem unsolvable. A3 means that a more preferred item is more likely to win a comparison. If it is not true, the meaning of "more preferred" becomes vague and we may not have enough information to recover the ranking. **7) Minor.** Thanks. We will make the introduction of the shorthand MNL more properly in the final version.

**Reviewer #3. Question 1. p1, p2)** (i) Theorem 2 is not a corollary of Theorem 3. Theorem 2 states an instance-wise lower bound for all $\delta$-correct algorithms and input instances satisfying A1-A3, while Theorem 3 only works for the MNL model. (ii) In Theorem 3, the algorithms know *a priori* that the input instances satisfy the MNL model, but in Theorem 2, the algorithms do not. Thus, Theorem 3 is also not a corollary of Theorem 2. **p3)** (i) Eqn(1) is no larger than Eqn(2). The right hand side of Eqn(1) (i.e., $\min\{...\}$) is no larger than $\sum_{i \in [n]} \tilde{\Delta}_i^{-2} \log n$ (by letting each $x_i = 1/n$). Summing up the left hand side of Eqn(1) (i.e., $\sum_{i \in [n]}[...]$) and $\sum_{i \in [n]} \tilde{\Delta}_i^{-2} \log n$, we get Eqn(2). Thus, Eqn(1) $\preceq$ Eqn(2). (ii) There are cases where Eqn(1) is strictly lower than Eqn(2). For example, if $\tilde{\Delta}_1 = n^{-2}$ and $\tilde{\Delta}_i = 0.1$ for all $i \geq 2$, then the right hand side of Eqn(1) is $\Omega(\tilde{\Delta}_1^{-2} + \sum_{i \geq 2} \tilde{\Delta}_i^{-2} \log n)$ (by letting $x_1 = 1/2$), lower than $\sum_{i \in [n]} \tilde{\Delta}_i^{-2} \log n$, making Eqn(1) $\prec$ Eqn(2). **Question 2.** (i) PLPAC-AMPR only works for the MNL model, so it only occurs in Figure 2(b) type-MNL. (ii) Active Ranking is for finding the Borda ranking [15]. For type-random instances, Borda ranking may not be the same as the exact ranking, so we do not compare Active Ranking in this case (so not present it in Figure 2(c) type-random). **Typos. p1)** Yes, we agree, and we will add the clarification in the final version. **p2)** The "2" near $O(...)$ in Line 87 is a footnote index, not "square". We will make it clearer in the final version.

[Meta-Review · NeurIPS 2019]

The paper considers a setting where one can actively query for pairwise comparisons (there is also an extension to listwise comparisons) and where the goal is to find the total ranking of the items. This is an important problem, in an area of significant interest to the NeurIPS community. The results are strong enough to recommend acceptance. However, there are also a number of misleading statements made in the paper that were flagged in the review process. It is very important that the authors fix these in the camera ready submission. (1) The abstract says "while most of the previous works either require prior knowledge or focus on approximate rankings". Due to this and other artifacts of writing of the paper, a reviewer incorrectly believed from reading this paper that "Active ranking under few assumptions is an important question, and previous works only solves this under the PAC setting" (as mentioned in the reviewer's initial review). The reviewer only later realized that [15] and others address exact ranking. (2) There was also confusion about the misleading statement "that Borda-Score algorithms are not suitable for exact ranking". The review team disagrees with such as statement as the assumptions in Borda-Score algorithms [15, 16, 21, 29] and this submission are simply incomparable. (3) The author's rebuttal stated that " If it is not true, the meaning of “more preferred” becomes vague and we may not have enough information to recover the ranking." That is incorrect as shown in the Borda-Score papers. (4) The paper considers the weak stochastic transitivity model in A3 for pairwise comparisons, and we suggest making a connection to this model to help the reader who are familiar with the literature (see arxiv 1510.05610 for an overview of different stochastically transitive models; also a good related read regarding these models is arxiv 1605.03933). The authors should carefully rewrite parts of the paper to remove all unclear and misleading statements that are discussed above and in the initial reviews.